# Energy Storage Performance of Polymer-Based Dielectric Composites with Two-Dimensional Fillers

**DOI:** 10.3390/nano13212842

**Published:** 2023-10-26

**Authors:** Liwen You, Benjamin Liu, Hongyang Hua, Hailong Jiang, Chuan Yin, Fei Wen

**Affiliations:** 1Faculty of Mathematical and Physical Sciences, University College London, London WC1E 6BT, UK; 2Environmental and Chemistry, Middlebury College, Middlebury, VT 05753, USA; 3Talent Program from China Association for Science and Technology and the Ministry of Education, Beijing Science Center, Beijing 100190, China; 4Department of Materials Science and Engineering, Boston University, Boston, MA 02215, USA; 5College of Electronics and Information, Hangzhou Dianzi University, Hangzhou 310018, China; 6School of Mechanical Engineering, Hangzhou Dianzi University, Hangzhou 310018, China

**Keywords:** dielectric property, nanocomposites, energy storage, 2D filler, breakdown strength

## Abstract

Dielectric capacitors have garnered significant attention in recent decades for their wide range of uses in contemporary electronic and electrical power systems. The integration of a high breakdown field polymer matrix with various types of fillers in dielectric polymer nanocomposites has attracted significant attention from both academic and commercial sectors. The energy storage performance is influenced by various essential factors, such as the choice of the polymer matrix, the filler type, the filler morphologies, the interfacial engineering, and the composite structure. However, their application is limited by their large amount of filler content, low energy densities, and low-temperature tolerance. Very recently, the utilization of two-dimensional (2D) materials has become prevalent across several disciplines due to their exceptional thermal, electrical, and mechanical characteristics. Compared with zero-dimensional (0D) and one-dimensional (1D) fillers, two-dimensional fillers are more effective in enhancing the dielectric and energy storage properties of polymer-based composites. The present review provides a comprehensive overview of 2D filler-based composites, encompassing a wide range of materials such as ceramics, metal oxides, carbon compounds, MXenes, clays, boron nitride, and others. In a general sense, the incorporation of 2D fillers into polymer nanocomposite dielectrics can result in a significant enhancement in the energy storage capability, even at low filler concentrations. The current challenges and future perspectives are also discussed.

## 1. Introduction

In recent years, there has been significant interest in the advancement of high-energy-density storage devices due to the escalating demand for renewable and sustainable energy sources and embedded integration technology. Electrical energy storage plays a key role in mobile electronic devices, stationary power systems, and hybrid electric vehicles (Figure 1) [1,2]. Dielectric energy storage stands out as a highly appealing and viable approach for energy storage and release when compared to alternative systems [3,4]. Dielectric materials possessing exceptional electrical, mechanical, and thermal properties play a crucial role as the primary facilitator in electrostatic energy storage devices, commonly referred to as dielectric capacitors. This is primarily due to their distinctive ability to generate ultra-high power density, exhibit low loss, and withstand high operating voltage [5]. The enhancement of dielectric performance and energy storage density has been a primary focus of numerous scientists and engineers in the field of energy storage research [2,6,7,8,9]. Materials with relatively high dielectric permittivity, low dielectric loss, high dielectric strength, low processing temperature, and high flexibility are highly needed for energy storage [10,11,12]. 

There is a substantial need for the development of new composites with superior electrical energy densities since current inorganic materials and organic materials fall significantly short of rising demands in advanced applications. Studying composites is motivated by the fact that diverse materials can be mixed to provide unique physical/chemical properties that are very different from those of the individual components [7,13]. Polymer-based 0–3 composites with diverse fillers are being explored for their improved dielectric properties, ease of manufacture, and flexibility. Nanofillers including ceramics, semiconductors, and conductors can boost nanocomposites’ dielectric characteristics and energy storage performances. In the last 5 years, many different reviews have given systematic summaries from different aspects to describe the roadmap and strategy of this field [4]. For 0–3 dielectric composites, there are five critical factors which can determine the film quality, dielectric properties, and the energy storage performance: (i) the selection of the polymer matrix, (ii) the type of the filler [14], (iii) morphologies and dimensions of the filler [15,16], (iv) interfacial engineering [9,17,18], and (v) the structure of the composite [19]. Among these critical factors, as is well known, filler morphologies and dimensions have a significant impact on how well composites operate as dielectric materials. In general, four types of fillers can be distinguished based on their morphologies: (1) zero-dimensional, 0D (e.g., nanoparticles, quantum dots); (2) one-dimensional, 1D (e.g., nanowires, nanofibers, and nanotubes); (3) two-dimensional, 2D (nanosheets, nanoplates, and nanoclays); and (4) three-dimensional, 3D (e.g., fille networks). It is widely believed that 2D fillers hold significant potential for dielectric composites due to their ability to generate multiple micro-lamellar structures within the composites. These structures offer two main advantages: (i) they facilitate the development of strong Maxwell–Wagner effects, thereby enhancing the polarization degree, and (ii) they effectively impede the expansion of breakdown routes, leading to improved breakdown strength and energy storage efficiency. In the last 10 years, polymer nanocomposites based on 2D nanomaterials have been widely studied. Figure 2 presents the trends in the number of articles with the keywords “dielectric” and “energy density,” and “dielectric” and “2D” or “dielectric” and “two dimensional” published in the refereed journals from 2012 to now.

In this review, the recent developments and strategies of polymer-based composites with 2D fillers are summarized (Figure 3). The fundamental and theoretical models for dielectric composites are first discussed in Section 2 to show how to predict the dielectric constant with the content of filler. Section 3 focuses on the selection of the polymer matrix and how it determines the performance of the composite. Section 4 summarizes the recent progress in achieving enhanced dielectric properties and energy storage capabilities of 2D polymer nanocomposites, including the structures and various 2D fillers. After that, a succinct summary and some potential outcomes are given.

## 2. Polymer-Based Dielectric Composites

### 2.1. Basic Information of Dielectric Energy Storage

The performance of a dielectric material is determined by the following parameters: dielectric permittivity (*ε_r_* or *k*), dielectric loss (tan *δ*), displacement–electric field relationship (*D*–*E*), and breakdown strength (*E_b_*) [10,11,12]. The energy stored in a dielectric material under an electric field *E* can be expressed by the shadow area in Figure 3, in which different relationships between *E* and *D* are presented as Equation (1):(1)$$U_{discharged}=U_{e}=\int E\cdot dD$$
where *U_e_* is the energy storage density, defined as the energy stored in a unit volume (J/m^3^). For linear dielectrics, it is well known that the energy density of a dielectric material is proportional to the product of permittivity and the square of the applied electric field, and can be expressed as Equation (2).
(2)$$U_{discharged}=\frac{1}{2}\varepsilon_{0}\varepsilon_{r}E^{2}$$
where *ε_0_* is the vacuum permittivity (8.85 × 10^−12^ F/m). Clearly, the dielectric properties of a material can effectively described by three crucial parameters: dielectric permittivity *ε_r_*, dielectric loss tan *δ*, and dielectric breakdown field *E_b_* applied on that material. Moreover, the schematic of the discharge energy efficiency *η* at high field is also shown in Figure 4. *η* is the ratio between the discharged energy and total energy and the loss can be calculated by 1 − *η* [3,4]
(3)$$U_{discharged}=\frac{1}{2}\varepsilon_{0}\varepsilon_{r}E^{2}$$

According to the electric conductivity of fillers, the polymer-based 0–3 composites can be classified into two types (Figure 5) [7,27]: (i) Dielectric–dielectric composites (DDC) consist of dielectric materials, particularly ferroelectric ceramics, as fillers. (ii) Conductor–dielectric composites (CDC) are composed of conducting materials, such as metals, carbon compounds, and conducting polymers, as fillers. In recent times, there has been a notable preference for semiconductive fillers as fillers in many applications. However, it is important to note that these fillers can still be classified into either the DDC or CDC category, depending on the dielectric performance of the composites as the filler content increases.

### 2.2. Dielectric–Dielectric Composites

Dielectric–dielectric composites are materials that combine dielectric particles or fillers with a polymer matrix and are specifically designed for their dielectric properties. These composites have shown promise in various energy storage applications, especially in the context of capacitors and energy storage devices that rely on dielectric materials to store electrical energy efficiently. For dielectric fundamental research and applications, it is of great interest to understand the dielectric response of a composite (or heterogeneous dielectrics) with different dielectric fillers. The dielectric properties of a heterogeneous dielectric are one of the earlier more interesting topics on the physics of dielectrics, which show different trends compared to one certain phase or neat polymer matrix [7,11]. Many mixing rules or models have been proposed to express or predict the effective dielectric permittivity (*ε_eff_*) of a system with two immiscible phases, especially for 0–3 polymer-based composites [4]. From the aspect of mathematical analogy, many of the models/formulas presented in this section for the calculation of the dielectric property are also valid for other physical properties, including the electric conductivity, heat conductivity, and diffusivity of such materials.

To explain and predict the effect of each phase on the composite dielectric properties, it is difficult to know the detailed information of the polarization response and electric field distribution under an external electric field [28]. It is also impossible to determine the dielectric response of a composite to the exact microstructure of the composite. Detailed information about the origination and derivation of each model/formula has been introduced in the previous review [7]. It should be mentioned that an extremely simplified model exhibiting the parallel connection (black curve) and series connection (red curve) of two phases has been proposed first [7]. For other models/formulas, the *ε_eff_* of the composite is expressed as a function of the composite’s composition (i.e., the content of fillers or volume fraction of fillers, *φ*), the dielectric permittivity *ε_m_* of the matrix, and the dielectric permittivity *ε_f_* of the filler materials. In some models/formulas, one other parameter related to the filler particles, such as shape or orientation, is also used. However, it is still difficult to simulate and predict the dielectric permittivity of a real composite. Some models are purely empirical, while others, like the Lewis–Nielsen equation and the Maxwell–Garnett equation, are appropriate for composites with a very low proportion of filler. Although some parameters have been involved in the equations and can fit some experimental results well, it is impossible to find one special formula for all composites. In recent studies, researchers typically plot some formulas which are close to the experimental data. 

Over the course of the past two decades, numerous ceramic–polymer nanocomposites have been investigated. This is especially true when high-k ferroelectric ceramics have been utilized, such as BaTiO_3_ [29,30,31,32,33], BaSrTiO_3_ [34,35,36,37], SrTiO_3_ [38], PbZrTiO_3_ [39], and CCTO [40,41,42]. Ceramic fillers with nanometer-scale dimensions have a number of advantages over fillers with micrometer-scale dimensions. These advantages include the ability to reduce the thickness of the composite while keeping its flexibility; improvement of the space charge production; increased voltage endurance; and prevention of partial discharge deterioration. The vast majority of ceramic nanoparticles come from commercial sources; however, some have been manufactured using solid state reactions, chemical procedures, or even by having their size reduced through the use of high-energy ball milling [43,44]. When it comes to questions about the storage of electrical energy, the high dielectric permittivity and robust electrical strength of the composite materials are both essential qualities to have. In order to avoid having a low electric breakdown field, the polymer matrix frequently has less filler.

### 2.3. Conductor–Dielectric Composites

The dielectric permittivity of a 0–3 conductor–dielectric composite cannot be explained by classical mixing rules but, instead, by the percolation theory [45]. Initially, when a modest amount of conducting filler is incorporated into a polymer matrix, the dielectric permittivity increases marginally with increasing filler concentration. Conductive particles are segregated and randomly dispersed in the matrix (Figure 6a), and the matrix dominates the associated electric characteristics of the composites. When the filler content is close to the critical volume fraction (Figure 6b), the dielectric permittivity is multiplied by hundreds/thousands, compared to that of the polymer matrix (Figure 6c). The critical volume fraction is the so-called percolation threshold (*φ_c_*), which is defined as the phase transition from the small, isolated particles to the interconnected channels [46]. When the concentration of conducting filler exceeds the critical concentration (*φ*_c_), the composite material exhibits conductivity. An additional augmentation of the filler content leads to the formation of a more extensive network of conductive channels within the composites, resulting in the establishment of a conductive skeleton (as depicted in Figure 6d). Consequently, the dielectric permittivity of the composites begins to exhibit a decline.

Many researchers are focusing on the concentration of the filler approaching the percolation threshold from the insulator regime (*φ* < *φ_c_*) since the dielectric permittivity undergoes a sharp rise to obtain the giant dielectric permittivity [47,48]. Divergence of dielectric permittivity around the insulator regime (*φ* < *φ_c_*) is caused by the formation of pure conducting channels through the whole composite. This can be thought of as a parallel link with an abnormally large capacitance. The strong nonlinearity of dielectric permittivity near the percolation threshold is caused by the large electric response in the thin barriers, which block off the conducting channels in the composite. For a random binary system if we assume the pure dielectric as a matrix, it was obtained as
(4)$$\varepsilon_{eff}\propto \varepsilon_{m}{\left(\varphi_{c}-\varphi \right)}^{-s}$$
where *φ < φ_c_* and *s* (>0) is a critical exponent [45,47]. Clarkson outlined two primary facets pertaining to the behavior of the dielectric permittivity in close proximity to the percolation transition. The first aspect entails a power–law relationship with respect to the volume fraction, while the second side involves a frequency-dependent behavior [49]. From some numerical simulations and results of static systems, it was concluded that *s* ≈ 0.7 [50]. Equation (4) is widely used in the literature to fit the experimental results [7,11,45]. Furthermore, Equation (4) is usually normalized as
(5)$$\varepsilon_{eff}=\varepsilon_{m}{\left(\frac{\varphi_{c}-\varphi }{\varphi_{c}}\right)}^{-s}$$

In Equation (5), the dielectric permittivity of the composite should be the dielectric permittivity of the matrix when there is no conducting filler added.

In recent decades, various conducting fillers have been selected, such as metal (Ag, Ni) [51,52], carbon materials (nanotubes, fibers, graphite, graphene) [53,54,55,56], conducting polymers [57,58,59,60,61], etc. The most important advantage is that using the conducting fillers can achieve a high dielectric constant and high energy density in the low filler content region, maintaining the high breakdown field and the mechanical performance of the composites [45,53,62]. Researchers also discovered that the fabrication processes used to create nanocomposites have a significant impact on the percolation threshold. Consequently, research has been investigated on many types of conducting fillers with a variety of morphologies, including nanospheres, nanotubes, nanobars, nanowires, and nanoplates. The implementation of a homogeneous dispersion of conductive fillers, along with the application of an insulating shell coating on such fillers, has the potential to somewhat mitigate the rise in dielectric loss. The fabrication of composites with high permittivity and low dielectric loss involves the preparation of metal particles that are coated with a core–shell structure consisting of metallic oxide fillers [63]. Carbon nanotubes (CNTs) are considered favorable options among the various conductive fillers due to their notable attributes, including strong electrical and thermal conductivity, as well as a significant aspect ratio [64,65]. One of the challenges associated with utilizing CNTs as fillers is the tendency for them to form agglomerates inside the polymer matrix. Modification and functionalization are widely utilized techniques for enhancing the dispersion of CNTs within host polymers [53,66]. Recently, researchers proposed 2D conducting fillers, including graphene, graphene oxide (GO), reduced graphene oxide (RGO), and MXene, to achieve a uniform dispersion of conductive fillers in the dielectric matrix to maximize the composite’s electrical properties [67], which will be discussed in Section 4.4 and Section 4.6.

In summary, conductor filler-based dielectric composites hold promise for energy storage applications, especially where a combination of high energy density, rapid discharge, and lightweight materials is required. However, addressing challenges related to stability and manufacturing complexity is crucial for their widespread adoption in various energy storage systems. Researchers and engineers continue to explore innovative approaches to enhance the performance of these composites.

### 2.4. Investigation on Polymer-Based Dielectric Composites

The investigation into polymer-based dielectric composites for energy storage is an exciting and multidisciplinary field that combines materials science, electrical engineering, and energy storage technologies [68,69]. Polymer-based dielectric composites have garnered significant interest due to their potential for high energy storage capabilities, lightweight nature, and ease of processing. As shown in Figure 7, for composites with 2D fillers, some key points must be considered during the investigation.

Polymer matrix. Polymer matrices play a crucial role in the study of dielectric composites for energy storage due to their ability to significantly enhance the overall performance and capabilities of such composites. Dielectric composites are materials composed of two or more distinct components with differing dielectric properties. When used for energy storage applications, these composites store electrical energy through the polarization of their dielectric materials in the presence of an electric field. Polymers can have excellent electrical insulating properties and good breakdown strength, which is the ability of a material to withstand high electric fields before experiencing electrical breakdown [70,71,72,73]. The polymer matrix helps to isolate and protect the embedded filler materials from high electric fields, contributing to the overall robustness of the composite. Detailed information on various polymer matrices will be discussed in Section 3.

Dimension of fillers. It is critical to investigate various filler materials that can be integrated into polymer matrices for the purpose of fabricating composites. As previously mentioned in Section 1, in addition to the polymer matrix and filler types, the size and shape of the filler particles significantly influenced the dielectric characteristics and energy storage performance. When comparing composites containing micro-sized ceramic fillers to those including nano-sized ceramic fillers, it is generally observed that the latter displays a more uniform microstructure. Additionally, these composites tend to have lower dielectric permittivity, while they also exhibit a significantly lower loss and larger breakdown field. As a result, composites with nano-sized ceramic fillers are considered more suited for applications related to energy storage. In general, the classification of fillers can be categorized into four distinct types. The materials under consideration include spherical particles with zero-dimensional (0D) characteristics, wires or fibers with one-dimensional (1D) properties, sheets or platelets with two-dimensional (2D) attributes, and clusters or network structures with three-dimensional (3D) characteristics [74,75]. Based on the principles of the effective medium theory, it can be posited that the dielectric permittivity is influenced by the depolarization factor, which exhibits a significant reliance on the aspect ratios of the ceramic fillers present inside the composite material. Composites, including fillers with larger aspect ratios, demonstrate an increased dielectric permittivity. Consequently, composites utilizing 1D fillers at lower concentrations may exhibit comparable dielectric permittivity to composites containing 0D fillers. This, in turn, leads to a higher breakdown field and, subsequently, a higher energy density [39]. Furthermore, the presence of small, specialized surfaces on the fillers with a large aspect ratio led to a drop in surface energy and a reduction in the extent of particle aggregation inside the polymer matrix. The aspect ratio of 2D fillers is higher compared to 1D fillers, leading to an augmentation in both the polarization density and interfacial polarization within the conductor–insulator system [76]. 

Microstructure and morphology. The microstructure and morphology of polymer-based dielectric composites play a crucial role in determining the electrical, mechanical, and thermal properties of composites. Achieving a uniform dispersion and distribution of fillers within the polymer matrix is crucial for optimizing the dielectric properties [77]. Agglomerations or clusters of fillers can lead to local variations in dielectric constant and hinder charge movement, reducing the overall effectiveness of the composite. The interactions between the filler particles and the polymer matrix at the interface play a vital role in determining the overall performance of the composite [17,18]. Good interfacial adhesion ensures effective stress transfer between the matrix and the fillers, leading to improved mechanical properties. The detailed discussion will be shown in Section 4.1.

Characterization and performance. Proper characterization can achieve the desired performance of the composites. Dielectric measurement is to show the material’s ability to store electric charge under an applied electric field. Higher dielectric constants are desirable for energy storage applications as they indicate a higher charge storage capacity. A lower dielectric loss indicates a higher energy storage efficiency and less heat generation. It is also important to characterize how the material’s performance changes across the frequency spectrum and temperature range relevant to the intended application [40,48,78]. Breakdown strength is the maximum electric field that the composite can withstand before breakdown occurs. A higher breakdown strength is crucial to prevent catastrophic failure at high voltages. In addition, the polarization under different electric fields and the efficiency and time of the charge–discharge processing are also critical to evaluate the energy storage capability. 

## 3. Dielectric Polymers

Polymer capacitors are more attractive for energy storage applications because they are inexpensive and possess a high dielectric strength, high temperature stability, and easy processing. As discussed in the introduction, a high dielectric strength plays a critical role in achieving high energy density. In addition, for polymer-based composites, the polymer matrix usually determines the dielectric properties, thermal properties, and energy storage ability of proposed composites [71,79,80]. In this section, different types of dielectric polymers are discussed, and their dielectric and energy storage performance are summarized in Table 1.

### 3.1. Non-Ferroelectric Polymers for High-Temperature Film Capacitors

The D–E relationship of a linear dielectric polymer is almost linear, so there is no polarization hysteresis loss, which is also called non-polar polymers (Figure 8), including epoxy, polycarbonate (PC), polyethylene (PE), polyethylene terephthalate (PET), polyimide (PI), polyester (PS), poly(methyl methacrylate) (PMMA), polypropylene (PP), polyphenylene sulfide (PPS), poly(vinyl chloride) (PVC), polyurethanes (PU), polytetrafluoroethylene (PTFE), etc. [4,70,79,81]. Polyester and polyimide offer a reasonable dielectric permittivity and have high operating temperatures. The disadvantage is their relatively high dissipation factor, which increases with temperature and frequency [82]. The beneficial properties of polypropylene are due to the polypropylene chain molecules, which do not have polar groups. Polypropylene has a higher breakdown voltage than other non-polar polymers [82]. The typical energy density achievable with polypropylene film at room temperature is 1.2 J/cm^3^. A metallized biaxially oriented polypropylene (BOPP) was prepared with a dielectric permittivity around 2.2 and energy density of 2.4 J/cm^3^ [82]. Ho et al. approached BOPP by polymerization of the monomer by UV light [83]. The energy density reached 5 J/cm^3^ and the breakdown strength increased 5% from 650 MV/m. Chung and coworkers synthesized a family of cross-linked polypropylene thin film. The high breakdown strength (650 MV/m) and energy storage capacity (5 J/cm^3^) was obtained due to the cross-linking effect [84]. Nevertheless, it has been observed that the conduction loss becomes more prominent at higher applied fields in numerous commonly utilized linear dielectrics, such as BOPP. Qing Wang’s group focused on polymers of which the dielectric properties exhibit considerable stability over a large frequency and temperature range, such as poly(ether ether ketone) (PEEK) and poly(phthalazinone ether ketone) (PPEK) [85,86]. Both of the polymers showed a higher dielectric permittivity (3–4), higher breakdown strengths (>400 MV/m), and higher energy density (>3 J/cm^3^) compared to many other high-performance polymers, indicating that they are promising candidates for high-temperature capacitor applications.

As discussed in the previous section, BOPP exhibited a low dielectric permittivity and low operating temperature due to its low melting temperature (<140 °C). Some alternative dielectric polymers, such as polycarbonate (PC), poly(ethylene terephthalate) (PET), and poly(phenylene sulfide) (PPS), can work at higher temperatures (>125 °C) but still suffered from their low dielectric permittivity [93]. It is necessary to synthesize a polymer possessing two criteria: the ability to (i) present as strongly dipolar to enhance the dielectric permittivity and (ii) make dipoles follow the applied field easily to avoid high loss [94]. Recently, some polar polymers or functionalized polymers have been fabricated to increase the dipole moment, eliminate the polarization hysteresis loss, and achieve high-energy-density storage, especially for high-temperature film capacitors [95]. The polarization mechanism in strongly dipolar polymer materials is primarily governed by orientation polarization. This phenomenon can be effectively characterized by the Frohlich model, which takes into account the short-range interaction between molecules and the resulting deformation polarizations [89]. As illustrated in Figure 8, compared to linear polymers, the modified polymers with an introduced polar group exhibit a larger dielectric constant and higher energy density. Researchers reported a series of polar polymers that have a very-high-energy-density storage, such as aromatic polyuria (ArPU), aromatic polythiourea (ArPTU) [87,88,96], meta-aromatic polyuria (meta-PU) [94,97], poly(arylene ether urea) (PEEU) [90], and poly(acrylonitrile butadiene styrene) (ABS) [98]. For example, meta-PU was synthesized via a green route which modified the molecular structure in the polyurea systems by controlling the dipolar density and dipole moment. A high storage electrical energy density of 13 J/cm^3^ and efficiency of 91% can be achieved at 670 MV/m. Compared to ArPU and ArPTU, meta-PU has a higher dipole moment and higher dipole volume density at the same electrical field. Poly(arylene ether urea) (PEEU) was synthesized via replacing the CH_2_ group in ArPU by the more polar ether group, which resulted in an increased dielectric permittivity of 4.7 and energy density of 13 J/cm^3^. Most importantly, PEEU exhibited excellent thermal stability up to 250 °C and discharged energy density of 9 J/cm^3^ at 120 °C. Recently, Zhang et al. synthesized a new family of ion-containing poly(4-methyl-1-pentene) (PMP) copolymers [91,99], which have a high thermal stability (up to 160 °C), a high energy density (35 J/cm^3^) at a high breakdown strength (1300 MV/m), and a high charge/discharge energy efficiency (>90%). The remarkable synergy between the high energy density and low dielectric loss in zwitterions-grafted copolymers can be attributed to the covalent bonding that restricts ion polarization and the effective charge trapping facilitated by the zwitterions, as previously demonstrated. The present study introduces an innovative approach to attain a substantial energy density and mitigate dielectric loss in polymer dielectrics.

### 3.2. Ferroelectric Polymers and Blend Polymer Matrix

Ferroelectric polymers have been widely used in various electronic applications, including actuators, acoustic transducers, and artificial muscles [100,101]. Poly(vinylidene fluoride) (PVDF) was the first electroactive polymer and is commonly chosen as a polymer matrix, it has been studied intensively in last 20 years [2,4,102,103]. It exhibits good mechanical and electric properties, such as piezoelectricity and ferroelectricity which exhibits both significant piezoelectric and ferroelectric properties. In this section, a summary on the energy storage behavior of ferroelectric polymers is given, including pure PVDF, PVDF-based copolymers, and PVDF-based terpolymers [10,104]. 

Zhang’s group studied the energy storage of PVDF in the α, β, and γ forms [105]. The presence of a high remnant polarization in β-PVDF is attributed to the D–E loop, which is defined by an all-trans polymer chain configuration. The observed disparity in remnant polarization between γ-PVDF and α-PVDF may be attributed to the TTTG conformation, which exhibits a greater level of polarity compared to the TGTG chain conformation. The energy density (1.5 J/cm^3^) and loss under low electric fields of PVDF in its three crystal forms have been shown to be identical. The γ phase samples have a maximum discharged energy-storage density of 14 J/cm^3^ due to their ability to withstand the highest breakdown field of 500 MV/m. Thakur et al. developed a novel approach to alter the surface properties, which can enhance the performance of energy storage systems and find applications in many fields [106]. A green aqueous functionalization of PVDF through dopamine is used to significantly enhance the dielectric properties, which results in a novel material for possible applications. 

A number of (PVDF)-based polymers have been studied many years, such as poly(vinylidene Fluoride-Trifluoroethylene) [P(VDF-TrFE)], poly(vinylidene fluoride-chlorotrifluoroethylene) [P(VDF-CTFE)], poly(vinylidene fluoride-hexafluoropropylene) [P(VDF-HFP)], and P(VDF-TrFE-CFE) terpolymers [2,4,102,103]. The presence of hysteresis in ferroelectric materials can be attributed to the energy barrier encountered during the reversal of the polarization direction. Ferroelectric polymers exhibit significantly elevated levels of losses as a result of the delayed switching of dipoles in response to the alternating electric field. Hence, it may be concluded that the P(VDF-TrFE) copolymer is unsuitable for the purpose of electrical energy storage. Guan et al. reported that the P(VDF-TrFE) 93/7 copolymer can have an energy density around 18 J/cm^3^ under 350 MV/m [107]. It was found that a high energy irradiation can introduce defects, such as chain scission, the formation of double bonds, etc. The irradiated P(VDF-TrFE) behaves like a relaxor, with a transition temperature much lower than unirradiated copolymers [108,109].

In order to enhance the characteristics of electrical energy storage, it is imperative to address two crucial concerns. The first pertains to the mitigation of dielectric and ferroelectric losses within ferroelectric polymers. The second involves gaining a comprehension of the underlying dipole reorientation and switching mechanism in the reaction to an alternating electric field. One potential strategy for confinement involves modifying conventional ferroelectrics to exhibit relaxor-like ferroelectric behavior, characterized by smaller hysteresis loops. Extensive research has been conducted on a novel copolymer, P(VDF-CTFE), which exhibits a remarkable combination of high electric energy density and discharge speed. The energy density of this copolymer has been shown to reach impressive levels, ranging from 17 to 25 J/cm^3^, as investigated by numerous research organizations [110,111]. It was first studied in 2006 by Q. M. Zhang’s group that a highly recoverable energy storage density of 17 J/cm^3^ was realized in P(VDF-CTEE) 91/9 mol.% at 575 MV/m [112]. Zhou et al. reported that a higher density of *U_e_*~25 J/cm^3^ can be obtained under a breakdown field >700 MV/m by further improving the film processing conditions and copolymer film quality. The films were prepared from a standard extrusion-blown process at 190–250 °C [110]. Q. Wang’s group studied the effect of the crystal structure on the polarization reversal and energy storage of P(VDF-CTFE) [111]. With different quenching and annealing temperatures, different crystallines can be obtained with different breakdown fields. Different crystallite structures can affect the dipole orientation behavior, which plays a vital role in determining and improving the high-field dielectric performance. Another PVDF-based copolymer with a higher electrical breakdown field is P(VDF-HFP) [107,113,114]. Zhou et al. achieved an electrical energy density that was higher than 25 J/cm^3^ of P(VDF-HFP) 95.5/4.5 at room temperature, which represents a one order of magnitude improvement over the widely used state-of-the-art BOPP capacitor films [113]. Wang Qing’s group obtained an energy density around 27 J/cm^3^ of P(VDF-HFP) 96/4 [107]. They also studied the energy storage of P(VDF-HFP) films with different crystal orientations by using different preparation and processing methods [114]. As shown in Figure 9, P(VDF-HFP) films with multiply treatments exhibited larger energy density at the same electric field compared to PVDF or P(VDF-CTFE) polymers.

Aside from that, there are many groups that proposed the use of a blend polymer matrix (BPM), mixing PVDF-based ferroelectric polymers with other linear dielectric polymers. Extensive research has been conducted on both ferroelectric relaxation and conduction at high fields in order to address the issue of substantial energy dissipation in ferroelectric polymers based on PVDF. Subsequent studies have revealed that the restriction of relaxation-induced high loss can be achieved through the grafting or combination of mixable polymers, specifically poly(methyl methacrylate) (PMMA) and polystyrene (PS). For example, the integration of mixable polymethyl methacrylate (PMMA) into copolymers based on polyvinylidene fluoride (PVDF) has noted to diminish dipole alignment along the electric field. This adjustment may not significantly affect the discharged energy density, but it could substantially elevate it [121,122,123]. Most notably, the crystal phase transition from the desirable phase to the undesirable phase of PVDF that is generated by a high electric field can be avoided with the use of PMMA. As a result, PMMA has seen widespread use as a material for the modification of PVDF-based copolymers in the production of energy storage capacitors. Thermoplastic polymers have the characteristic of being able to undergo melting and subsequent processing, making them suitable for utilization in polymer film capacitors as these capacitors are currently fabricated using processes that are compatible with the melt processability of thermoplastic polymers. The efficacy of crosslinking thermoplastic polymers in enhancing their physical properties has been well established. The enhancement of various properties, such as hardness, stiffness, wear and impact resistance, thermal stability, and insulation strength, can be observed when PE or PMMA undergoes crosslinking, resulting in the formation of crosslinked polyethylene. This material has proven to be highly effective as an insulation material in high-voltage power systems. Furthermore, it has been observed that crosslinking enhances the tensile stress, breakdown strength, and capacitive performance of ferroelectric polymers [124].

In summary of this section, the large-scale fabrication of ultra-thin, high-temperature dielectric films of a high quality will be the future trend for all organic dielectric capacitors. How to obtain high power densities, a fast charge–discharge speed, and great stability at elevated temperatures (>150 °C) is the main task in this research field. On the one hand, controlling the exponential increase in electrical conduction with temperatures will contain the degrades of the capacitive performance at elevated temperatures [125]. On the other hand, new ultra-thin film fabrication technology is needed to improve the quality of the film because the performance of dielectrics highly rely on the fabrication methods and environmental conduction [126]. These pioneer works will lead the future work for the scale-up preparation of ultra-thin dielectric films and the minimization of capacitors under extreme environments.

## 4. Polymer-Based Composites with 2D Fillers

Recently, 2D fillers have exhibited superior potential in improving the energy storage performance of polymer nanocomposites [127] (Figure 3), including low-*k* 2D nanofillers (e.g., montmorillonite nanosheets) [128,129], high-*k* 2D nanofillers (e.g., BaTiO_3_, TiO_2_, and NaNbO_3_) [20,130,131], and other graphene-like 2D layered nanomaterials (i.e., molybdenum disulfide, MXene, and hexagonal boron nitride) [132]. During the process of breakdown, it is possible to conceptualize 2D fillers as conductive barriers that have the ability to restrict the movement of charge and impede the progression of electrical treeing. Additionally, the utilization of 2D materials possessing a significant specific surface area has the potential to facilitate the dispersion of fillers in a direction perpendicular to the applied electric field. This dispersion mechanism, in turn, contributes to the mitigation of the local electric field gradient, ultimately resulting in an improvement in the breakdown strength of dielectric polymer composites. In this section, polymer-based composites with various 2D fillers are discussed and summarized.

### 4.1. Structures of the Composites

Polymer-based dielectric composites for energy storage can come in various structural configurations, including single-layer, sandwich, and multilayer arrangements (Figure 10) [15,19,133]. Each configuration has its own advantages and is suited for specific applications. For 2D fillers, the structure is critical to the energy storage performance due to the geometry of the 2D filler itself [67].

In a single-layer structure, the composite consists of a homogeneous mixture of the polymer matrix and filler materials. The filler materials are dispersed throughout the polymer matrix, creating a uniform material with enhanced dielectric properties. This structure is relatively simple to fabricate and is suitable for applications where a moderate increase in energy storage capacity is required. The most widely used fabrication methods are solution casting and spinning coating, followed by thermal treatment [25]. The interface between the polymer matrix and filler materials is critical for the performance of the composite [19]. A well-adhered and uniform interface ensures efficient charge transfer and reduced internal electric field concentrations. This leads to enhanced dielectric properties and breakdown strength. The coupling agent plays a role in improving this interface by promoting adhesion and compatibility [5]. A coupling agent is often used to improve the compatibility between the polymer matrix and the fille materials. It helps to enhance the adhesion and interaction at the matrix–filler interface, resulting in improved overall properties of the composite. Coupling agents can be silanes or other surface-modifying agents that have functional groups that are compatible with both the polymer and the filler, including the -OH group [134], dopamine [135], metal oxides [136], PM7F [137], and other agents.

The sandwich structure involves two different designs: one is to place a layer of the composite with 2D fillers between two layers of polymer layers (0-x-0), and the other one is to place a layer of the polymer matrix between two layers of composites with 2D fillers (x-0-x), as shown in Figure 10b [138]. Here, 0 means 0 vol.% of the filler and x means x vol.% of the 2D filler in the matrix. Recently, many researchers have investigated both structures and made a comparison between them [139,140]. The multilayered structure means the composite is composed of more than three layers. The utilization of multilayer structural materials has emerged as a potential approach for enhancing the breakdown strength [141], with a particular emphasis on the development of high energy storage performance dielectrics. This strategy has proven to be crucial in advancing the exploitation of such materials [77].

### 4.2. High-k 2D Platelets and Nanosheets

#### 4.2.1. Lead-Free Ceramics

For dielectric composites as 2D ceramic fillers, extensive research has been conducted on lead-free piezoelectric ceramics due to their environmental tolerance, biocompatibility, and ease of fabrication [35,142,143,144]. Furthermore, these ceramics demonstrate a comparatively lower permittivity in comparison to ceramics containing lead. This characteristic offers several advantages: (i) it mitigates the dielectric mismatch between the fillers and the matrix, (ii) it eliminates any remnant polarization, and (iii) it extends the breakdown path while preventing charge migration. These factors collectively contribute to the enhancement of the breakdown strength. The utilization of these additives presents a fresh methodology for augmenting the energy storage density of composite films under conditions of comparatively modest electric fields. The investigation of composites containing 2D ceramic fillers has received considerably less attention in studies compared to composites with 1D ceramic fillers. This disparity can be attributed to the restricted commercial accessibility and challenges associated with the manufacturing of composites incorporating 2D ceramic fillers. Researchers discovered that the dielectric permittivity and disintegration strength can be simultaneously enhanced by adding a small amount of 2D filler, resulting in multiple attempts. According to the findings of these studies, the geometry of the 2D fillers played a significant role in the concomitant enhancement. When using 2D fillers with a high dielectric permittivity, it is desirable to obtain a high energy density at a relatively low electric field, given these achievements.

The most popular high-*k* and lead-free materials are BaTiO_3_ [145], SrTiO_3_ [146,147], Ba/SrTiO_3_ [137,148], and BiFeO_3_ [149]. Wen et al. reported BT/PVDF composites using 2D platelets, which were prepared and investigated in this work (Figure 11a–d) [20]. The composites were fabricated using a straightforward method, including a solution casting and quenching treatment, without the use of a multiple-layer architecture or surface chemical treatment. The composite exhibited notable enhancements in both dielectric characteristics and energy storage capability in comparison to the original PVDF material. The composite film, which contained a little amount of BT (1 wt.%), demonstrated a significant discharge energy density of 9.7 J/cm^3^ at an electric field strength of 450 MV/m. This value is twice as high as that of pure PVDF and nearly five times higher than the discharge energy density of the leading commercially available BOPP film. Moreover, composite films exhibit exceptional cycle stability and fatigue resistance. The simulation also provided insights into the local electric field and local polarization distribution of the composite material. The energy performance of the material in question is comparable to or exceeds that of several previously documented composites containing 0D BT particles, 1D BT nanowires, and two other 2D dielectric fillers [150]. The dielectric characteristics and energy storage capacity of composites consisting of 2D ceramic platelets were explored by D. Zhang and colleagues through fabrication and experimentation. For example, (Na_0.5_Bi_0.5_)_0.93_Ba_0.07_TiO_3_ (NBBT) platelets, with a size of 5 µm and thickness of 0.2–0.5 µm, were used as the filler in a P(VDF-HFP) polymer matrix. By combining different routes, the film exhibited a high discharged energy density of 10 J/cm^3^ at 258 MV/mm [151]. The researchers also conducted an investigation on a nanocomposite consisting of P(VDF-TrFE-CTFE) and BaTiO_3_ platelets. This nanocomposite was generated using a three-step molten salt process and was subsequently modified using a unique RAFT polymerization method [145]. The composites containing 15 vol.% BaTiO_3_ platelets achieved an energy density of 1.26 J/cm^3^ under the influence of an electric field of 60 kV/mm. Additionally, the researchers incorporated two distinct components, namely graphene sheets and BaTiO_3_ platelets, into the PVDF polymer matrix. This resulted in the achievement of a significantly high permittivity value of 66.2, while simultaneously attaining an exceptionally low dielectric loss of 0.048 [152]. Gao’s group investigated the influence of plate-like (Ba_0.6_Sr_0.4_)TiO_3_ on the dielectric properties of a PVDF polymer [137]. The findings of the study indicated that the plate-like particles had a favorable alignment and consistent directionality within the PVDF matrix. The obtained values for the energy storage density were 6.36 J/cm^3^, while the dielectric loss was found to be 0.042, indicating an ultra-low level.

Besides the BaTiO_3_-based ceramics, other types of lead-free ceramics, such as NaNbO_3_ (NN) and K_0.5_Na_0.5_NbO_3_ (KNN), are also excellent lead-free ferroelectric ceramics with relatively high dielectric constants of ∼300 at 1 kHz [153]. Combined with the trilayered architecture, the 2D NN/PVDF composites illustrate a high discharge energy density of 13.5 J/cm^3^ at 400 MV/m (Figure 11e–h) [131], and P(VDF-HFP) composites with core–shell NaNbO_3_@Al_2_O_3_ platelets obtained an enhanced discharge energy density of 14.59 J/cm^3^ at 400 MV/m [136]. This achievement can be ascribed to the utilization of high-aspect ratio 2D NN/KNN platelets, as well as the development of trilayered architecture composite films. These films consist of two outer layers made of high dielectric constant NN/PVDF, while the middle layer is composed of high-breakdown-strength pristine PVDF. This finding is further supported by the investigation of another set of KNN/PVDF composites [154]. In addition, bismuth-layered structures, like Na_0.5_Bi_4.5_Ti_4_O_15_ [135], and SrBi_4_Ti_4_O_15_ [155], which easily form plate-like particles from the molten salt method, have also been used as the filler because of their relatively low dielectric permittivity. The reduction in dielectric mismatch between the filler and the polymer matrix leads to a decrease in the distortion of the electric field at the interface, resulting in an improvement in the electric breakdown strength. An exemplary composite material consisting of 2D SrBi_4_Ti_4_O_15_ nanosheets demonstrated a high energy storage density of 11.69 J/cm^3^ and a discharge efficiency of 78.94% [155].

#### 4.2.2. Perovskite Nanosheets

Besides the non-polar characteristic of the layered ferroelectric materials (e.g., CaBi_4_Ti_4_O_15_, SrBi_4_Ti_4_O_15_) discussed in Section 4.2.1, another type of 2D perovskite oxide nanosheets (Ca_2_Nb_3_O_10_) have recently introduced a new platform for dielectric energy storage [156,157,158]. Few studies have examined its application as efficient fillers for polymer nanocomposite capacitors, which are more commonly employed as all-inorganic free-standing dielectric capacitors [159]. When the total number of free electrons hits a certain level, a complete dielectric breakdown takes place. Therefore, reducing the kinetic energy of energetic electrons is one way to raise the *E_b_* while suppressing the quantity of secondary electrons. Increasing the breakdown strength of polymer nanocomposites is a major problem in achieving a high energy density and good dependability under high voltages. By including negatively charged Ca_2_Nb_3_O_10_ nanosheets with a thickness of approximately 1.5 nm (Figure 12a), Bao et al. hypothesized that they might considerably increase their breakdown strength and energy storage and, thus, obtained PVDF-based nanocomposite capacitors which exhibited the highest energy density (36.2 J/cm^3^) and a significantly improved breakdown strength (792 MV/m) among all flexible polymer-based dielectrics (Figure 12b,c) [21]. The same significant improvements in the breakdown strength and energy density of polystyrene-based nanocomposites serve as proof that the method is generalizable. Phase-field simulations show that the local electric field created by the negatively charged Ca_2_Nb_3_O_10_ nanosheets sandwiching the positively charged polyethyleneimine is responsible for the further enhanced breakdown strength. This local electric field suppresses the secondary impact-ionized electrons and obstructs the breakdown path in nanocomposites. The outcomes indicate a brand-new potential of flexible capacitors with high energy densities. In addition, Shen et al. fabricated a polymer nanocomposite, P(VDF-HFP)/Ca_2_Nb_3_O_10_, with a storage energy of 35.9 J/cm^3^, which mainly benefitted from an improved breakdown strength of 853 MV/m [160]. Meanwhile, to assess the energy storage capacity, a machine learning technique was planned in this work (Figure 12d). They discovered that the breakdown strength of polymer nanocomposites could be greatly enhanced by the parallel perovskite nanosheet’s preference for blocking and then driving charges to migrate along with the interfaces in the x-y plane. Besides the Ca_2_Nb_3_O_10_, another group synthesized a novel ferroelectric Sr_2_Nb_2_O_7_ nanosheet via a simple two-step hydro-thermal reaction to generate an increase in polarization [121]. A significantly improved Weibull breakdown strength of 602.5 MV/m and a resulting high discharge energy density of 28.39 J/cm^3^ was obtained. Very recently, besides CNO nanosheets and multilayer structures, interfacial engineering, which uses bidirectional-matched aluminum oxide interface transition regions between polyimide and CNO nanosheets, was proposed [161]. The successful creation of the bm-interface was demonstrated by a multitude of experimental and characterization results. These findings indicate that the interface has an inherent ability to hinder the movement of carriers and the conduction of electricity in the composites. The composites exhibited exceptional energy storage capabilities in both conventional and high-temperature situations.

#### 4.2.3. Metal Oxides

From an energy storage standpoint, the integration of nanoparticles, with permittivity values in the range of hundreds or even thousands, into polymers, which typically exhibit permittivity values below ten, may not be advantageous in terms of attaining a substantial enhancement in energy density [162]. Due to the significant disparity in permittivity between the filler and the polymer matrix, the primary factor contributing to the augmented dielectric permittivity is the heightened average field within the polymer matrix, while the filler phase retains a minimal amount of stored energy. Moreover, the existence of a substantial disparity in permittivity between the two phases gives rise to a profoundly non-uniform electric field, leading to the formation of a composite material with a much reduced effective breakdown strength. Due to these reasons, many metal oxides have been used as fillers to improve the energy storage performance, including TiO_2_ [162,163,164,165], ZrO_2_ [166,167,168], Al_2_O_3_ [169], SiO_2_ [170], Fe_3_O_4_ [171], ZnO [172,173,174], and SnO_2_ [175].

Although many articles have reported dielectric composites with metal oxides, there are still limited publications on 2D metal oxides. Sheng et al. successfully exfoliated a ZrO_2_ nanosheet from ZrClO_2_·8H_2_O powders through a simplified process and fabricated PVDF/ZrO_2_ nanocomposites [176]. The reduced leakage current and increased Young’s modulus of the composites achieved a much improved energy density of 11.03 J/cm^3^ at a breakdown strength of 519 MV/m. After that, 2D titanium dioxide (TiO_2_) nanosheets [130] and monolayer titania (Ti_0_._87_O_2_) nanosheets [22] were introduced as the filler separately, in 2018 (Table 2). Both of these composites exhibited an improved breakdown strength (>570 MV/m) and large efficiency (>60%). Especially, it can obtain an energy density of 21.1 J/cm^3^ in a single layer of PVDF with titania monolayers, which represents an enhancement of 1758% over the BOPP (1.2 J/cm^3^ at 640 MV/m), as shown in Figure 13a–d. For high-temperature capacitor applications, the alumina, as a representative linear dielectric material, features stable chemical and thermal properties, a large band gap (~9 eV), and an excellent insulation performance [177]. Wang’s group systematically studied the influence of varied morphologies of Al_2_O_3_ fillers (nanoparticles, nanowires, and nanoplates) on the dielectric performance [178]. Besides the advantages of the metal oxides, the increased insulation is initially ascribed to the orientation of 2D metal oxide nanosheets along the in-plane directions perpendicular to the external electric field, which results in an increase in path tortuosity in the electrical treeing process (Figure 13e–g). When polymer chains are bonded to metal oxide nanosheets, their mobility is diminished, which inhibits the transfer of charge carriers through the polymer chains’ loose mobility. The increased insulation contributes to both the increased failure strength and the reduced dielectric loss.

**Table 1 nanomaterials-13-02842-t001:** The parameters of energy storage of polymers.

Polymer	Preparation or Treatment	Thickness	*ε_r_*	tan *δ*	*E_b_*	*U_e_*	*P_m_*	Ref.
		(μm)			(MV/m)	(J/cm^3^)	(C/m^2^)	
A: Non-Ferroelectric polymer								
Polypropylene (PP)	-	10	2.2	<0.02	640	2.4	-	[82]
Polyester (PET)	-	3	3.3	<0.5	570	1–1.5	-	[82]
Polycarbonate (PC)	-		2.8	<0.15	528	0.5–1	-	[82]
Polyphenylenesulfide (PPS)	-		3.0	<0.03	550	1–1.5	-	[82]
Biaxially oriented polypropylene (BOPP)	metallized and UV irradiation	7	2.2	-	720	5	-	[83]
Poly(propylene-co-p(3-butenyl)styrene)	cross-linking reaction	10	3.0	-	650	>5	0.013	[84]
Polyimide (PI)	spin-coated	2.5	3.4	-	300	1–1.5	-	[81]
Poly(ether ketone ketone)	melt pressing under pressure	25	3.6	0.003	450	3	0.008	[85]
Poly(phthalazinone ether ketone) (PPEK)	chemical reaction/hot pressing	20–40	3.5	<0.01	450	3.9	-	[86]
Aromatic polyurea	thermal vapor deposition and annealing	2.5	4.2	0.005	800	>12	0.035	[87]
Aromatic polythiourea	microwave-assisted polycondensation	1–5	4.5	<0.002	1000	22	0.045	[88]
Meta-phenylene polyurea (m-phPU)	polycondensation	1–5	5.7	0.017	1000	>20	0.038	[89]
Poly(arylene ether urea) (PEEU)	thermal polycondensation	-	4.7	0.008	700	13	0.035	[90]
Modified poly(4-methyl-1-pentene)	Ziegler–Natta and hot press and stretch	10	5.0	0.015	612	>7	0.027	[91]
B: PVDF based co-, tri-polymer								
PVDF			12		590	2.4		[82]
PVDF	quench γ-PVDF	20–30			500	14	0.09	[105]
PVDF	dopamine modified	20–30	32	<0.002	140	2.7	-	[106]
P(VDF-TrFE) 93/7		80–100	12.6	<0.01	350	18	0.11	[107]
P(VDF-CTFE) 91/9	extrusion at 190–250 °C	10	13	0.03	600	25	0.13	[110]
P(VDF-CTFE)	quench in liquid N_2_ then anneal at 25 °C	15–25			400	8	0.065	[111]
P(VDF-CTFE)	cooling to R.T then anneal at 110 °C	15–25			500	10	0.08	[111]
P(VDF-HFP) 96/4	solution cast and uniaxial stretching	8		<0.01	600	27	0.08	[107]
P(VDF-HFP) 95.5/4.5	extrusion and stretch at 110 °C	3–11	12	-	700	>25	-	[113]
P(VDF-HFP)	solution cast at RT	15	5.6	0.07	550	20	0.07	[114]
P(VDF-HFP)	solution cast and stretching and annealing	8	9.6	<0.05	550	22	0.085	[114]
P(VDF-HFP)	melt-pressing and quench and stretching	20	12.2	0.03	500	20	0.07	[114]
P(VDF-TrFE-CFE) 63/37/7.5	suspension polymerization	10–15	50	<0.2	400	9	0.09	[115]
P(VDF-TrFE-CTFE) 88.0/5.2/6.8	Direct polymerization	20	10	-	500	10.3	0.086	[116]
P(VDF-TrFE-CTFE) 65.6/26.7/7.7		30–40	60	-	>500	>13	0.1	[117]
C: Ferroelectric polymers–polymer composites								
P(VDF-TrFE-CTFE)-g-PS (14 wt.%)	hot press at 240 °C and quench and stretched	80–100	9	<0.01	500	21	0.08	[107]
P(VDF-CTFE)97/3-g-PS (34 wt.%)	hot press and quench and stretch	80–100	5	0.006	600	10	0.025	[179]
P(VDF-TrFE-CTFE)80/18/2-g-PEMA (22 wt.%)	quench at 0 °C	20	6.5	<0.05	550	14	0.075	[118]
P(VDF-CTFE)91/90-BA (10%)	ultraviolet radiation		-	-	400	22.5	0.12	[119]
PC/PVDF multilayers 50/50		0.38/12	3/12	<0.03	600	11	-	[180]
PVDF/PMMA (40 wt.%)	quench at 0 °C	20	6	0.05	400	6	-	[181]
α-VDF oligomer-P(VDF) 80/20	uniaxial pressure to eliminate defects	2	4.9	-	868	27.3	0.162	[120]

**Table 2 nanomaterials-13-02842-t002:** The dielectric properties and energy storage performance of composites with 2D fillers.

2D Filler	Polymer	Size of Filler	Thickness (µm)	Coupling Agent	Structure	Content	*ε_r_*	tan *δ*	*E_b_* (MV/m)	*U_e_* (J/cm^3^)	*η* (%)	Ref.
High-k ceramics												
BaTiO_3_	P(VDF-TrFE-CTFE)	L: 10 μm, T: 10 μm	-	PM7F	Single layer	15%	90.2	0.1	60	1.26	74.2	[145]
BaTiO_3_	PVDF	R: 3–8 μm, T: 0.2–0.5 μm	10	-	Single layer	0.3%	11.9	<0.04	450	9.7	55	[20]
SrTiO_3_	PVDF	R: 3–15 μm, T: 0.2–0.3 μm	-	dopamine	Single layer	1 wt.%	10.66	-	357	9.48	57.2	[146]
Ba_0_._6_Sr_0_._4_TiO_3_	PVDF	R: 3–8 μm, T: 0.1 μm	100	PM7F	Single layer	40%	62.2	0.042	29	6.36	-	[137]
NaBiBaTiO_3_	P(VDF–HFP)	L: 5 μm, T: 0.2–0.5 μm	10	PVP	30-1-1-1-30	1–30%	25.3.	0.05	258	14.95	90	[151]
NaNbO_3_	PVDF	L: 2–5 μm, T: 0.1–0.5 μm	15	PDA	3-0-3	3%	11	~0.04	400	13.5	71	[131]
NaNbO_3_	P(VDF-HFP)	L: 1–5 μm, T: 0.1–0.5 μm	10	Al_2_O_3_	Single layer	3%	12	<0.05	440	14.59	70.1	[136]
K_0_._5_Na_0_._5_NbO_3_	PVDF	L: 17–40 μm, T: 0.4–3.5 μm	20	-	0-3-0	3%	12	<0.05	350	14.5	80.2	[154]
Na_0_._5_Bi_4_._5_Ti_4_O_15_	PVDF	L: 15–20 μm	-	dopamine	Single layer	1 wt.%	16	0.1	300	9.45	52.3	[135]
SrBi_4_Ti_4_O_15_	PVDF	R: 1 μm, T: 0.25 μm	15	-	0-5-0	5%	13	<0.05	385	11.69	78.9	[155]
Ca_2_Nb_3_O_10_	PVDF	L: 150 nm, T: 1.5 nm	12	-	Single layer	2.1 wt.%	10.5	-	792	36.2	61.2	[21]
Ca_2_Nb_3_O_10_	P(VDF-HFP)	L: 37.4 nm, T: 3 nm	-	-	Single layer	0.1%	-	-	853	35.9	-	[160]
Ca_2_Nb_3_O_10_	PVDF	L: 100 nm, T: 1.7 nm	11	-	Sandwich		11	<0.05	710	25.1	80	[182]
Sr_2_Nb_2_O_7_	PVDF	L: 35 nm, T: 3 nm	9	-	Single layer	5 wt.%	11	<0.05	600	28.4	71	[121]
Metal oxides												
ZrO_2_	PVDF	L: 20–40 μm, T: 20 nm	-	-	Single layer	1 wt.%	10	<0.04	519	11.03	67.4	[176]
Ti_0_._87_O_2_	PVDF	L: 15–20 μm, T: 1.2 nm	10	-	Single layer	1 wt.%	12	<0.03	650	21.1	60	[22]
TiO_2_	PMMA/P(VDF-HFP)	R: ~200 μm, T: 6 nm	10	dopamine	Single layer	5 wt.%	10	~0.04	570	13.0	63	[130]
Montmorillonite												
MMT	PVDF	-	20	-	Single layer	0.2 wt.%	28	0.032	873	24.9	>60	[24]
Na^+^/MMT	PVDF	-	30	ionic liquid	Single layer	-	15	<0.02	100	5.5	81	[183]
Na^+^/MMT	polypropylene	T: 20–25 nm	-	-	Single layer	0.4 wt.%	3.75	<0.01	530	5.2	94.9	[184]
MMT	PVDF	-	15	-					460	7.26	69	[185]
Transition metal dichalcogenides												
MoS_2_	PVDF	D: 1–2 µm	-	-	Single layer	0.4%	11.3	0.07	200	2.3	~72	[186]
Bi_2_Te_3_	PVDF	R: 0.4–1 µm, T: 0.1 µm	-	Al_2_O_3_	Single layer	10 vol.%	140	0.05	50	-	-	[76]
Bi_2_Te_3_	P(VDF-HFP)	R: 0.4–1 µm, T: 0.1 µm	-	SiO_2_	Single layer	10 vol.%	70.3	0.058	<500	-	-	[187]
MoS_2_	PI	L: 1 µm	18	-	Single layer	1 vol.%	3.3	<0.02	395	3.35	>80	[188]
MoS_2_	Chitin	L: 2 µm	15	-	Single layer	5 wt.%	~9.8	~0.025	350	4.91	>80	[189]
MoS_2_	g-PMMA/PI	T: 1–2 µm		MMA	Single layer	3 wt.%	4.2	0.015	450	8.6	61.7	[190]
MoS_2_	P(VDF-CTFE-DB)	R: 3–5 µm, T: 0.2–0.5 µm	15	ZnO	Single layer	2 wt.%	12.9	0.047	300	7.2	83	[25]
Graphene-based fillers												
Graphene	P(VDF-TrFE-CFE)	L: 0.1–0.4 µm, T: 1.6 nm	~20	HBPE-g-HFBA	Single layer	0.1 wt.%	~15	~0.04	250	5.0	78.1	[191]
Graphene	P(VDF-CTFE)	L: 0.2–0.6 µm, T: 1.3 nm	12	HBPE-g-PTFEMA	Single layer	0.8 vol.%	24.8	0.06	250	4.6	62	[192]
GO	P(VDF-HFP)	T: 1 nm	29	-	Sandwich	2 wt.%	~11	~0.1	300	10	77	[193]
BNNS												
BNNS	P(VDF-TrFE-CFE)	L: 0.4 µm, T: 10–70 nm	-	-	Single layer	12 wt.%	38	0.03	650	20.3	78	[194]
BNNS	PMMA	L: 0.4 µm, T: 2 nm	-	-	Single layer	12 wt.%	~3.6	0.044	473	3.5	86	[195]
BNNS	PVDF	L: 0.5–1 µm, T: 2–10 nm	30	-OH	Single layer	6 wt.%	11.1	~0.014	517	13.1	-	[134]
BNNS	PVDF	L: 1–2 µm, T: 2 nm	10	-	Single layer	8 wt.%	8.3	<0.05	486	7.25	-	[196]
BNNS	PVDF	L: <3 µm, T: 3.9 nm	12	-	Sandwiched	0.16 vol.%	~11	<0.07	612	14.3	73	[197]
B_16_-BN	PVDF	L: 1 µm, T: 1.5–2.5 nm	25	-OH	Single layer	8 wt.%	9.6	<0.03	436	9.8	-	[198]
BNNS	PEI	L: 1 µm, T: 2.7 nm	10	-hydroxyl	Single layer	4 vol.%	~3.3	<0.02	700	7.67	93.6	[199]
h-BN	P(VDF-CTFE)	L: ~0.4 µm, T: 1.5 nm	15	-	Sandwiched	0.4-0-0.4	35.1	<0.03	300	9.1	62.8	[140]
BNNS	P(VDF-TrFE-CFE)	-	5–10	NH_2_/Epoxy	Matrix free	18 wt.%	~32	<0.05	742	31.8	72.7	[23]
BNNS	P(VDF−HFP)/PMMA	L: ~0.1–0.2 µm, T: 5 nm	25	lysozyme	Single layer	5 wt.%	~10	<0.06	500	14.9	71	[122]
BNNS	Cellulose	L: 0.6 µm, T: 1.3 nm	-	-	Single layer	10 wt.%	~7	0.02	370	4.1	75	[200]
BNNS	Cellulose	-	-	–COO^−^	Single layer	4 vol.%	~8	<0.03	384	3.9	66	[201]
BNNS	Chitin	L: ~0.5 µm, T: 3.3 nm	15	-	Single layer	6 wt.%	7.1	0.018	450	8.7	90	[202]
Ti_3_C_2_T_X_ (MXene)												
Ti_3_C_2_T_X_	PVDF	T: 2–4 µm	45	-	multilayer	2-1-0.1	20	0.04	350	12.5	>60	[203]
Ti_3_C_2_T_X_	PVDF	-	22	-	multilayer	4:5	41	0.028	300	7.4		[204]
Ti_3_C_2_T_X_	PI	-	5	-	Single layer	0.5 wt.%	3	<0.02	648	8.67	84.1	[26]

**Figure 13 nanomaterials-13-02842-f013:**
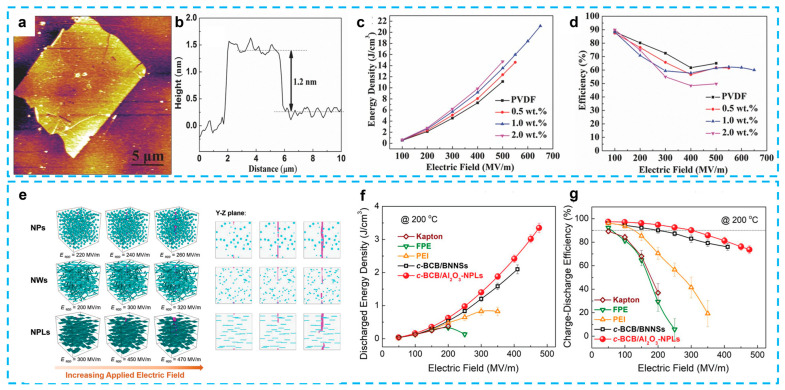
Nanocomposite capacitors with significantly enhanced energy density and breakdown strength utilizing a small loading of monolayer titania. (**a**) AFM image and (**b**) the corresponding height profiles of TOMLs. (**c**) energy density calculated from D−E loops, and (**d**) charge–discharge efficiency of PVDF/TOML nanocomposites with different filler contents as a function of the electric field. Reproduced with permission from [22], Copyright 2017, WILEY-VCH. Scalable polymer nanocomposites with record high-temperature capacitive performance enabled by rationally designed nanostructured inorganic fillers. (**e**) The predicted breakdown path evolution. (**f**) Discharged energy density and (**g**) charge–discharge efficiency of high-temperature dielectric polymers and the c-BCB nanocomposites measured at 200 °C. Reproduced with permission from [178], Copyright 2019, WILEY-VCH.

### 4.3. Montmorillonite

Layered silicates (such as montmorillonite and laponite clays) have been employed as fillers to make polymer-based composites with superior qualities due to their abundance in nature, low cost, distinctive lamellar structure, high cation exchange, water-swelling capacity, and adaptable interlayer spacing [205]. Montmorillonite (MMT) is the most common silicate, which consists of many layers packed parallel to one other to create lamellae of around 1 nm in thickness and several micrometers in length. It indicates that each lamella has a high length-to-thickness ratio, which enables the passage of energy from the inorganic to the organic phase and back. Many studies have demonstrated that polymer nanocomposite materials have improved mechanical and dielectric characteristics [206,207,208], especially to improve the electroactive phase, enhance the polarization, and increase the tensile strength of PVDF-based polymers [209,210,211].

Recently, a research group examined the improvement of the energy storage capability because they found that adding MMT is beneficial to the breakdown strength of PVDF nanocomposites because of the superior mechanical reinforcement effect and nanometric interfaces of clay minerals, resulting in a higher polarization, improved dielectric performance, and breakdown strength [24,185,212,213,214]. Ghosh et al. made the discovery that intercalation could occur in PVDF/clay nanocomposites without the need for chemical treatment. This phenomenon was attributed to the vulnerability of the long hydrocarbon chain ligands typically employed in surface modification. The low dielectric constant of these ligands, in contrast to the high dielectric constant of the PVDF matrix, rendered them susceptible to the applied voltage, facilitating intercalation [24] (Figure 14a–d). The composites exhibited superior dielectric properties and great energy storage performance (24.9 J/cm^3^ at 873 MV/m). This work’s simplification and extensibility provide a cost-effective approach to achieving better *E_b_* and *U_e_*. These characteristics provide a realistic design for flexible and transparent nano-dielectric materials with a high dielectric response and exceptional energy storage performance.

Although many investigations have been performed, it was found that the improvement of the dielectric properties was limited by (i) the poor compatibility between the MMT and the polymer matrix, and (ii) the weak polarization of MMT. Some researchers believed that ion and liquid modification of MMT could improve the performance. Consequently, different ions, such as Li^+^- and Na^+^-modified MMT, were prepared and the related dielectric nanocomposites were investigated [183,184,215]. In addition, ionic liquid as an addition, along with ion modification, played a critical role in enhancing the energy storage properties (Figure 14e–g) [183]. For example, the PP-g-MAH nanocomposite film with an optimized org-MMT content of 0.4 wt.% possessed an excellent discharged energy density of 5.21 J/cm^3^ under 530 MV/m with a high efficiency of 94.9% [184]. Due to the interfacial chain movement constraint of PP-g-MAH, the incorporation of org-MMT somewhat decreased the nanocomposite’s dielectric constant. Alternatively, org-MMT increased the nanocomposite’s tensile strength by inhibiting the growth of electrical trees.

In summary, MMT as a low-cost and easy-processing filler is a good candidate compared to other 2D fillers. The mechanisms behind the improved energy storage capability are that (i) the interaction between PVDF-based polymers and MMT may decrease the mobility of the PVDF chains, thereby inhibiting charge transport through the loose amorphous region, and (ii) the MMT can perform as an insulating barrier to prevent current conduction in the PVDF matrix, thereby inhibiting charger carrier mobility and reducing leakage current.

### 4.4. Graphene-Based Nanosheets

The primary focus of research in the field of dielectric materials revolves around enhancing the dielectric constant and breakdown strength, while concurrently upholding a high level of charge/discharge efficiency. Hence, it is imperative to augment the mechanical characteristics of the polymer material while simultaneously preserving its electrical insulating capabilities, specifically its low dielectric loss. In the context of these applications, the incorporation of conductive two-dimensional carbon fillers, such as graphene, has been found to enhance mechanical performance. However, it has been observed that the inclusion of these fillers has a negative impact on dielectric characteristics. Graphite can be exfoliated to produce graphene, which is a single two-dimensional sheet. It has exceptional qualities, including being incredibly light, mechanically robust, thin, electrically conductive, and very strong. Graphene has, so far, found wide-ranging uses in the electronics, pharmaceuticals, composites, coatings, sensors, and energy industries. Graphene was employed as a conductive filler in the production of polymer-based nanocomposites with elevated permittivity, owing to the interfacial polarization that occurs between the fillers and the matrix. Liu et al. employed the technique of in situ polymerization to fabricate materials with a high dielectric permittivity. The topic of interest pertains to the study and development of nanocomposites comprising graphene and polyimide materials [216]. The dielectric permittivity had a significant enhancement with the augmentation of graphene content. Fan et al. fabricated nanocomposites consisting of graphene and PVDF with a multi-layered architecture [217]. The nanocomposites exhibited a remarkably low percolation threshold of 0.0018 in terms of volume percent of graphene, which stands as the most minimal value documented among PVDF-based nanocomposites. At a graphene volume percentage of 0.00177, the nanocomposites exhibited a significantly elevated dielectric permittivity of 340 at a frequency of 100 Hz. Upon surpassing the percolation threshold, the dielectric permittivity experienced a subsequent increase, reaching a substantial value of 7940 at a frequency of 100 Hz. However, it is noteworthy that this increase in dielectric permittivity was accompanied by a significant rise in dielectric loss. The study conducted by Wang et al. presents a noteworthy investigation whereby a three-dimensional aerogel was employed as a template for the synthesis of graphene/poly(vinyl alcohol) nanocomposites [218]. The inclusion of poly(vinyl alcohol) as a barrier in graphene aerogel/poly(vinyl alcohol) nanocomposites effectively prevents direct contact between the conductive skeletons. This results in a notable improvement in the dielectric properties of the composites. Specifically, the dielectric permittivity of these nanocomposites can reach a value as high as 1059, while the dielectric loss is remarkably low, measuring only 0.08. In a separate study, Zhang and colleagues conducted an emulsion polymerization process to graft varying quantities of polystyrene (PS) onto reduced graphene oxide (rGO-PS). Subsequently, they incorporated the resulting rGO-PS into a PS matrix, thereby producing nanocomposites of PS/rGO-PS [219].

However, compared to nanoparticles with the same loading, 2D nanosheets could percolate more easily, which is why there were limited reports on the energy storage of the composites with graphene-based nanosheets. Some groups used fluoro-HBPE as a polymer stabilizer to achieve a homogeneous distribution of graphene and reduced graphene oxide (rGO) and obtain an energy density around 5 J/cm^3^ at a low electric field (~250 MV/m) [191,192]. In many recent works, researchers usually mix graphene or rGO with the second filler and then introduced this in the polymer matrix.

### 4.5. Boron Nitride Nanosheet Fillers

Boron nitride nanosheets (BNNS), isolated from hexagonal boron nitride (h-BN) and referred to as “white graphene”, have drawn much attention because of their similar structure to graphene. BNNS have been widely used in a wide range of applications, from the fields of automotive, aerospace, healthcare and medical, and energy storage to electrical engineering [220,221]. BNNS have a wide band gap of (~5.6 eV) and excellent electrical insulation properties, including a high bulk resistivity (10^13^ Ω·cm) and low dielectric loss, which are proving to be particularly promising in terms of an enhanced electric field, increased energy density, and improved dielectric reliability [222,223]. In addition, under extreme conditions at high temperatures, most capacitors have a low energy storage efficiency and reduced energy density, so it is of great importance to develop capacitors with high charge and discharge efficiencies at high temperatures. It was found that BN nanomaterials have a superior fracture strength (165 GPa), high Young’s modulus (0.8 Tpa), high thermal stability (800 °C in the air), excellent coefficient of thermal expansion (−2.72 × 10^−6^ K^−1^), and outstanding thermal conductivity (300–2000 W·m^−1^·K^−1^) [224]. Recently, the introduction of BNNS exfoliated from BN particles into different polymer matrixes has been demonstrated to achieve an outstanding dielectric performance and ultra-high energy density, especially for high-temperature applications [225,226].

BNNS have been composited with different polymers for dielectric energy storage materials, such as PVDF [227], P(VDF-TrFE-CTFE) [194], crosslinked bisbenzocyclobutene (c-BCB) [228], poly(methyl methacrylate) (PMMA) [195], cellulose [200], polyimide [226], and polyetherimide (PEI) [199]. In contrast to conventional polymer nanocomposites, which typically require organic functionalization of the inorganic filler’s surface to achieve a uniform distribution within the organic phase, it has been observed that the polar nature of B-N bonds facilitates the dispersion of BNNS in polar organic solvents and polymer matrices characterized by high polarity, such as fluoropolymers and biopolymers. The preparation of BNNS typically involves the utilization of h-BN using the solution exfoliation technique. In the usual procedure, a quantity of 1 g of hexagonal boron nitride (h-BN) will be evenly distributed within a volume of 100 milliliters of N,N-dimethylformamide (DMF) with intense stirring, followed by sonication. The upper transparent layer was thereafter subjected to centrifugation and afterwards dried under vacuum conditions in order to acquire the precipitated product of BNNS [194,195]. For example, Li et al. used the solution exfoliation method to produce P(VDF-TrFE-CFE)/BNNS and PMMA/BNNS nanocomposites, respectively, which could be applied at high temperatures [194,195]. The thickness of BNNS obtained by solution exfoliation is 2 nm and the transverse dimension is 400 nm. It was found that the uniform dispersion of BNNS on the PMMA surface greatly improved the thermal conductivity and capacitance of the nanocomposites. The experimental results show that the discharged energy density can reach up to 20.3 J/cm^3^ from the 15 mm thick terpolymer nanocomposite with 12 wt.% of BNNS at 650 MV/m in P(VDF-TrFE-CFE)/BNNS composites, and 3.5 J/cm^3^ at 473 MV/m with 86% efficiency in PMMA/BNNS composites. Chen et al., using the same method, prepared PEI nanocomposite films containing two-dimensional hydroxyl-functionalized boron nitride nanosheets (h-BNNS) [199]. The experiments utilized polyetherimide (PEI) with a high glass transition temperature (Tg), moderate dielectric constant (*ε_r_*~3.2), low loss, and good mechanical strength as a polymer matrix. The nanocomposite films exhibit a great breakdown strength (*E_b_*~700 MV/m) at room temperature, high discharge energy density (~7.67 J/cm^3^), and high discharge efficiency (~93.6%). In addition, the nanocomposites exhibit excellent thermal stability at 500 MV/m and 150 °C, with a discharge energy density of 3.43 J/cm^3^.

Very recently, Huang’s group proposed a matrix-free method to efficiently reduce the effects of electron multiplication while improving the mechanical modulus and thermal conductivity of the polymer [23]. This method involves chemically attaching an amino-containing polymer to BNNS surfaces to form an electron barrier layer. The nanocomposites were able to significantly improve the break-down strength and significantly reduce leakage current, which led to a striking rise in discharge energy density. As shown in Figure 15, an ultra-high energy density (31.8 J/cm^3^) at a high breakdown strength (*E_b_* = 742 MV/m) was achieved, exhibiting good repairability by reversible chemisorption. The rationale behind this phenomenon is twofold: (i) the surface electron barrier layers of the BNNS exhibit a pronounced repulsive and obstructive influence on high-energy electrons when the BNNS are oriented perpendicular to the electric field, and (ii) the interconnected molecular networks within the nanocomposites substantially enhance the Young’s modulus while preserving elongation at the break.

For the development of BNNS nanomaterials, however, there are still numerous concerns and obstacles. Examples include the low chemical reactivity of BNNS, its weak contact with the matrix polymer, and its inhomogeneous dispersion on the matrix polymer. The incorporation of inorganic fillers must be handled with care as the interface and flaws caused by incompatibility between two materials can significantly increase dielectric loss and degrade breakdown performance. The chemical alteration of the raw BNNS surface poses challenges in directly introducing an electron barrier layer due to its inert molecular composition. The most frequently employed technique involves the chemical modification of filler surfaces, which can be achieved through processes such as grafting organic surfactants or applying a thin layer of polymers onto the particle surfaces.

### 4.6. Transition Metal Dichalcogenides (TMDs)

Recently, unlike graphene, most interesting and exciting 2D material transition metals are dichalcogenides (TMDs), such as molybdenum disulfide (MoS_2_), tungsten disulfide (WS_2_), and bismuth telluride (Bi_2_Te_3_), which have been explored intensively with the aim of achieving a good performance for various kinds of applications [229,230,231]. The TMD sheets exhibit a 2D shape and an ultra-thin thickness, and they display distinct physical, chemical, and electrical properties in comparison to their bulk counterparts. These materials possess a high band gap of 1.8 eV, rendering them chemically stable and capable of modifying the dielectric constant under the influence of an external electric field. Despite these characteristics, they do not introduce electrical conductivity to the polymer matrix. Moreover, they have the potential to enhance mechanical properties such as the elastic modulus, strength, toughness, and fatigue resistance.

Elsik et al. studied the dielectric and mechanical properties of polymer composites with a small amount of MoS_2_ nanoplatelets [232]. The authors exfoliated bulk MoS_2_ into nanoflakes, which were then dispersed in epoxy polymers, and characterized the tensile and fracture properties of the prepared composites. The mechanical properties of the epoxy were effectively enhanced by loading very low (below 0.2% by weight) fractions of MoS_2_ nanoplatelets, demonstrating the significant potential of two-dimensional transition metal dihalides as reinforcing additives in polymer composites. Although it is the first work studying composites with TMD fillers, this work did not discuss the electric breakdown and energy storage. Jiang’s group first reported that MoS_2_ superstructures can significantly influence flexible ferroelectric polymer composites, and two important studies are exhibited in this work [186]. Similar to the behavior observed in 2D fillers, composites based on MoS_2_ superstructures also display electrical behavior that is reminiscent of percolation, wherein a notable increase in dielectric constant is observed in close proximity to the percolation threshold. Furthermore, when MoS_2_ superstructures are subjected to mild loading, the resulting composites demonstrate a remarkable ability to endure strong electric fields and display a substantial increase in electric polarization. As a consequence, there is a notable enhancement in the capacity for electrical energy storage. For example, at an electric field of 200 MV/m, the total stored energy density of the composites with 0.4% MoS_2_ flower are 4.1 and 2.3 J/cm^3^, respectively. Although the value is much lower compared with other composites with 2D fillers, it is a good start and provides more insight in this type of composite.

In recent years, many groups have also focused on composites with MoS_2_ fillers, mainly focusing on the surface modification [190], different matrixes (PI, Chitin) [188,189], and hybrid fillers with conducting fillers (PPy Aluminum flake, ZnO, etc.) [25,233,234,235,236]. Within the single-layered composites, as shown in Figure 16a–d, Li et al. synthesized the MoS_2_ nanosheets coated with a PMMA layer using the SEP method to fabricate MoS_2_-g-PMMA nanosheets [190]. The dielectric constant of MoS_2_-g-PMMA/PI (MPP-3%) reaches 4.2, which is 20% higher than that of a pristine PI film, while the energy density reaches 8.6 J/cm^3^ at room temperature and 3.92 J/cm^3^ at 150 °C, which is 40% higher than the highest energy density of a pristine PI film. This is a much better result than what researchers have found before for high-temperature capacitor applications. The MoS2-g-PMMA/PI-based nanocomposite has a lot of potential for use as a high-temperature capacitor. Chen et al. investigated chitinous/MoS_2_ nanocomposite dielectric films, using the biodegradable, renewable, and biocompatible chitin, which is a natural polymer with an extremely high annual production [189]. The researchers successfully dissolved chitin in a low-temperature freeze–thaw cycle using a new environmentally friendly solvent, aqueous KOH/urea, and the experimental results showed that chitin has great potential for dielectric energy storage applications [237]. The study used the same method to dissolve chitin and the results showed that the dielectric constant and breakdown strength of the chitin/MoS_2_ nanocomposite increased, while the dielectric loss remained low. At a content of 5 wt.%, the composite film achieved a charge/discharge efficiency of over 80% and a breakdown strength of 350 MV/m, resulting in a high discharge energy density of 4.91 J/cm^3^. Wen et al. prepared mixed semiconductor nanofillers with different ratios of molybdenum disulphide (2D) nanosheets and zinc oxide (0D) nanoparticles using a wet chemical route and ultrasonic mixing, as shown in Figure 16e–i [25]. The P(VDF-CTFE-DB)/ZnO@MoS_2_ nanocomposites containing 2 mol% filler on the study surface exhibited a high power density and excellent fatigue reliability. The hybrid fillers can effectively improve the dielectric properties, breakdown field, and energy storage properties. P/ZnO@MoS_2_ composites with a 2 wt.% have a high energy density (7.2 J/cm^3^), high power density (0.17 MW/cm^3^), and high charge/discharge efficiency (83%). In addition to MoS_2_, Bi_2_Te_3_ has also been investigated as a conductive nanofiller for enhancing the dielectric characteristics of nanocomposite films. This is due to the favorable electrical conductivity and significant aspect ratio exhibited by 2D hexagonal nanoplates of Bi_2_Te_3_. Cheng et al. coated the Bi_2_Te_3_ with SiO_2_ and Al_2_O_3_ to effectively improve the dielectric properties and energy storage performance [76,187]. In addition, similar to composites with BNNS fillers, composites with TMDs also have good thermal stability. The energy storage performance at elevated temperatures is summarized in Section 5.

In summary, the role of the TMDs’ filler, which can improve the dielectric constant and maintain the relatively low dielectric loss, is ascribed to the following aspects: (1) morphology—exfoliated nanoplates with large aspect ratios, (2) electric aspect—high band gap and a tunable dielectric constant, (3) breakdown—efficient conduction barriers limiting charge migration toward electrodes, and (4) mechanical aspect—enhanced reinforcing additives with good tensile and fracture properties. However, there are still some limitations and bottlenecks in the current research; as such, few studies have so far investigated the effect of MoS_2_ nanosheets on the dielectric properties of polymer composites. (1) It is difficult to exfoliate thinner two-dimensional layers, (2) the environmental pollution issue during the preparation of TMDs, and (3) there remains a low energy density and breakdown field compared to other 2D fillers. Two methods can be interpreted to enhance the breakdown field and energy storage. The first is to use an insulating layer of metal oxides between the TMD and polymer matrix to prevent the accumulation of charge carriers at the interface. The coated 2D fillers in the matrix operate as nucleating agents to encourage the creation of nonpolar crystals in the region of the fillers; as the mobility of dipoles is constrained in these crystals, the reorientation of dipoles becomes more challenging and requires a stronger electric field. Another potential approach involves employing many layers and high-aspect-ratio 2D fillers across these layers. This strategy aims to establish effective conduction barriers that restrict the movement of charges towards the electrodes and obscure the formation of electric trees during breakdown.

### 4.7. MXene

MXene nanomaterials with graphene-like structures and advantageous conductivity have been produced and studied [238,239]. MXenes are 2D transition metal carbides and/or carbonitrides and have received increasing attention from scholars since their discovery in 2011 [240]. The general chemical formula of MXenes is M_n+1_X_n_T_x_ (“M” stands for early transition metal, “X” is carbon and/or nitrogen, “T” is a surface termination, x is the number of termination groups and n = 1–3). Some typical surface functional groups (T_x_) are reported as O, OH, and F (Figure 17a). These surface terminations make the surface hydrophilic and molecularly polar so that MXenes mix well with aqueous solutions and a variety of polar organic solvents, such as dimethylformamide (DMF) and dimethylacetamide (DMAc) [241]. Subsequently, more than 30 MXenes were created, each exhibiting unique features based on different M and X elements and their respective ratios [242]. At the same time, two-dimensional MXene materials have great potential for batteries, capacitors, and electromagnetic shielding devices due to their high aspect ratio and high conductivity (Figure 17b) [243,244,245,246]. In dielectric research, Ti_3_C_2_T_x_ MXene nanosheets, which have a higher Young’s modulus and lower conductivity than graphene oxide nanosheets, have been used to enhance the breakdown characteristics of filler components [247].

Tu et al. first investigated the effect of Ti_3_C_2_T_x_ doping into PVDF polymer nanocomposites to prepare high-*k* polymer nanocomposites [248]. At a frequency of 1 kHz and with a loading of 10.7 wt.% MXene, the dielectric constant was measured to be 1424. Additionally, the dielectric loss was found to be 0.35. It is worth noting that this represents the highest dielectric constant achieved while maintaining a dielectric loss below one. The aforementioned group has successfully devised a technique to augment the dielectric constant of MXene/P(VDF-TrFE-CFE) composites by the manipulation of flake dimensions and the concentration of surface functional groups [249]. The dielectric permittivity of the composite, which incorporates a large area (4.5 μm) of MXene flakes, exhibits a notable increase, reaching a value as high as 10^5^ in close proximity to the percolation limit. Nevertheless, the magnitude of the loss is significantly elevated, surpassing a threshold greater than ten. To mitigate the loss, the establishment of a well-organized configuration of MXene within the polymer matrix can effectively optimize the dielectric constant while simultaneously minimizing the dielectric loss. In the case of PVA/10.0 wt.% MXene, it was shown that high dielectric constants of up to 3166 accompanied by a low dielectric loss of 0.09 were achieved. These values surpass the previously reported dielectric data for MXene/polymer nanocomposites, particularly when considering the frequency range [250]. Recently, many MXene-based dielectric composites have been reported, most of which focused on achieving an ultra-high dielectric constant with low filler loadings but near the percolation threshold, such as a dielectric constant of 439 with a loss of 0.53 in MXene/acrylic resin [251], a dielectric constant of 539 with a loss of 0.06 in MXene/P(VDF-HFP) [252], a dielectric constant of 11,800 with a loss of 1.31 in MXene/PVC [253], a dielectric constant of 23.7 with a loss of 0.11 in MXene/PDMS [254], and a dielectric constant of 82.1 with a loss of 0.2 in MXene/PVDF [255]. However, because the energy storage performance was mainly determined by the low loss and low conductivity, most of the study still focused on the high dielectric permittivity [256,257,258,259] instead of investigating the energy storage capacity.

Although most of the studies on MXene-based nanocomposites focused on a high dielectric permittivity, there were still some reports on the energy storage capability. It should be mentioned that, usually, a multilayer-structure is a critical design for obtaining a good performance from MXene-based nanocomposites. Feng et al. endeavored to achieve a delicate equilibrium between high-k and high breakdown characteristics, resulting in the development of a novel gradient sandwich structure of a MXene/fluoropolymer nanocomposite [203]. The simultaneous enhancement of high-k features and a high breakdown strength in the inhomogeneous multi-layered PVDF-based composite with a concentration gradient of MXene can be attributed to the improved interface polarization between adjacent sub-layer interfaces, in addition to the MXene/PVDF interface polarization and the interface barrier effect between the adjacent two sub-layers. The sandwich composite, in its original state, had a permittivity value of 26 at a frequency of 100 Hz. Additionally, it demonstrated a breakdown strength of 350 MV/m and an energy density (efficiency) of 12.5 J/cm^3^ at the aforementioned breakdown strength. In their study, Li et al. employed the layer-by-layer hot-press technique to obtain a significantly improved dielectric constant. This enhancement can be attributed to the interfacial polarization that occurs between the PVDF and MXene materials [204]. The application of MXene onto PVDF results in the formation of a multilayer film, which exhibits an expanded surface area that facilitates enhanced charge accumulation at the interfaces. This augmentation in charge accumulation contributes to an increase in the polarization of the multilayer film, known as the Maxwell–Wagner–Sillars (MWS) polarization. The 4MXene/5PVDF film that was artificially produced demonstrated a dielectric constant of 41, a minimal dielectric loss of 0.028, and a comparable breakdown strength of 284 MV/m. The observed outcomes in energy storage can be attributed to two factors: One notable effect of a stronger interface barrier between the adjacent sub-layers in a sandwich-structured composite is the enhancement of the composite’s high breakdown strength. This is achieved by effectively limiting the growth of electric trees throughout the entire thickness of the sandwich composite when subjected to a high applied field. Conversely, the polarization of the MWS (Metallically Conductive MXene-Wrapped Single-Walled Carbon Nanotubes) arises between the MXene and the polymer, as well as between the neighboring sub-layers, hence contributing to the enhanced high permittivity of the composite material. Very recently, research still tried to achieve a high energy density in a single-layer film. PI-based nanocomposites with two-dimensional alkylene oxide as a filler exhibited significantly enhanced capacitive properties at high temperatures, as shown in Figure 17. Yu et al. introduced a Ti3c2TX colloidal solution directly into the PI matrix using an in situ polymerization method, which greatly improved the breakdown strength [26]. The largest discharged energy density of 8.67 J/cm^3^ and efficiency of 84.1% are achieved at 648 kV/mm in 0.5-wt.% oxidized MXenes/PI nanocomposites at room temperature (Figure 17c,d). The incorporation of layered filler in composites results in an enhanced interface between the polymer and the filler, which leads to the accumulation of charges at the interface, subsequently giving rise to a pronounced interfacial polarization. Consequently, the ongoing escalation of the interface results in the dominance of interface polarization, leading to an enhancement of the dielectric constant. It is plausible that the charge accumulated on the surface of oxidized MXene is effectively taken and dispersed by MXene possessing a lower Fermi level. This study helps to further investigate the preparation of high-performance dielectric polymer-based composites in a wide temperature range.

In summary, the fatal leakage current and conduction losses of composites are caused by the extremely high conductivity of MXene, which is not suitable for energy storage. However, there are still three strategies to make this combination more suitable for energy storage, including introducing ceramics fillers, using a multi-layer structure, and oxidation of the MXene surface. The purpose of all of these methods is to reduce the leakage current and increase the breakdown strength to enhance the energy density of polymer-based composites.

## 5. Concluding Remarks and Outlook

Significant progress has been achieved in the field of polymer-based dielectric composites and ultra-thin 2D material research during the past decade, including a wide range of investigations, from fundamental scientific inquiries to the development of state-of-the-art technological applications. The integration of 2D filler materials with energy storage applications has made notable progress, highlighting the influential role that dimensionality plays in shaping the fundamental characteristics of nanomaterials and their diverse array of practical uses. In this comprehensive review, we have systematically classified the latest advancements in the field of research under consideration. These advancements have been categorized based on several key aspects, such as the fundamental dielectrics involved, the significant influence of the polymer matrix, the diverse range of 2D fillers utilized, the methods employed for dielectric characterizations, the energy storage performances observed, and the promising potential applications that have garnered attention. However, there are still some limiting factors that hinder them in practical applications.

Firstly, the preparation of various 2D fillers is lacking. Various advanced synthetic processes can be employed to produce ultra-thin 2D nanomaterials, each possessing their own distinct advantages and limitations. The exciting aspect lies in the capacity to produce ultra-thin 2D nanomaterials that possess diverse structural attributes, including size, thickness, crystallinity, crystal phase, defect, doping, strain, and surface property. These properties offer significant advantages across a wide range of applications. Nevertheless, the discovery and production of ideal 2D fillers, particularly heterogeneous nanofillers or 2D core–shell fillers, may be hindered until a versatile and efficient method for fabricating free-standing 2D materials is developed.

Secondly, the practical application of 2D material polymer composites is hindered by the significant constraint of high dielectric loss resulting from their raised dielectric constants. The occurrence of dielectric loss in high-k materials is frequently attributed to several factors, including significant polarization, sluggish relaxation, the presence of conducting channels, as well as strain and stress, especially caused from graphene structure-based 2D fillers. The aforementioned paradox is frequently encountered in 2D polymer nanocomposites to a greater degree in comparison to their polymer equivalents. The preservation or enhancement of the charge–discharge efficiency in nanocomposites is associated with the insulating properties of these nanofillers. These nanofillers act as insulating barriers, impeding current conduction and minimizing leakage. The intriguing phenomenon exhibited by nanocomposites consisting of insulating 2D fillers seems to have resolved the paradox discussed before. Nevertheless, further investigations are required to comprehensively analyze and quantify these discoveries. The core–shell construction technique, while demonstrating enhanced efficiency, is often characterized by time-consuming processes, limited yields, and high costs. Furthermore, considering that the dielectric properties of the matrix are generally negligible due to the substantial influence of conducting fillers on composites, it is possible to modify several types of polymer matrices in order to attain a strong compatibility with fillers.

Thirdly, another significant challenge encountered in the field of creating 2D polymer nanocomposites pertains to the intricate nature of interactions occurring at the interface between the polymer and nanofiller. Additionally, there is limited comprehension regarding the impact of various factors such as size, shape, edge effects, and nanofiller volume on these interactions. For instance, in the case of composites, the inclusion of 2D nanofillers between polymer layers seems to exhibit superior dielectric characteristics compared to composites where 2D nanosheets are scattered isotopically. Composites containing 2D nanosheets or nanoplates have a superior performance compared to those incorporating nanoparticles. While the in-plane polarization generated by the form has been proposed as the cause, additional research is necessary to comprehend the impact of shapes prior to their utilization in actual contexts. The complexity of interface science increases when many polymers and various 2D materials are present as theoretical investigations typically focus on a single polymer and a single 2D material.

In summary, the achievement of high energy storage applications requires careful attention to design concerns since it involves finding a delicate balance between permittivity, dielectric loss, and breakdown strength. Solving the above challenges is essential for both fundamental science and practical applications. It is anticipated that these challenges, in conjunction with recent significant advancements in high-performance polymers and 2D materials, as well as ongoing fundamental research on dielectric phenomena, will likely result in the creation of scalable, high-performance dielectric materials for the design of energy storage devices.

## Figures and Tables

**Figure 1 nanomaterials-13-02842-f001:**
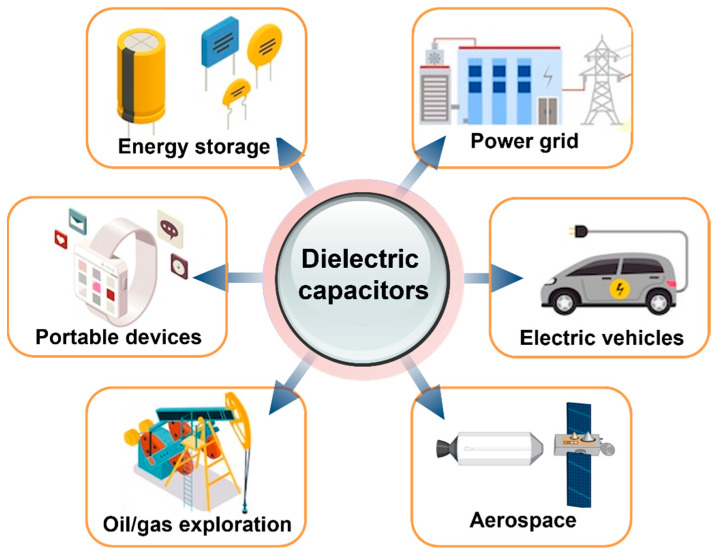
Applications of dielectric capacitors. Graphic elements are designed by freepik.com (accessed on 1 September 2023).

**Figure 2 nanomaterials-13-02842-f002:**
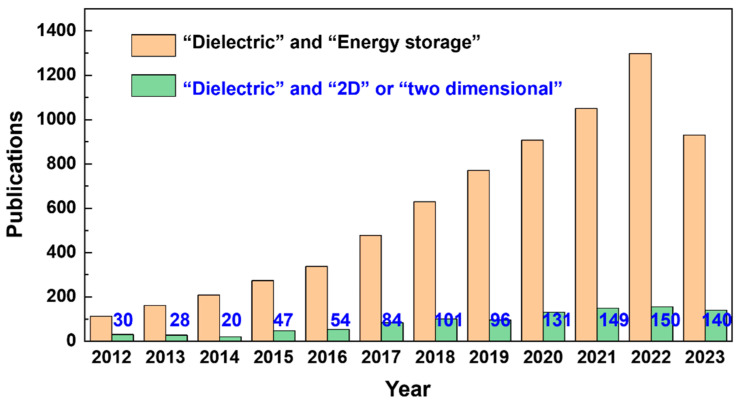
Trends in the number of articles on energy storage dielectrics published in the refereed journals from 2012 to 2023. The results were collected from Web of Science Core Collection using the keywords “dielectric” and “energy storage,” “dielectric” and “2D”, and “dielectric” and “two dimensional”, respectively.

**Figure 3 nanomaterials-13-02842-f003:**
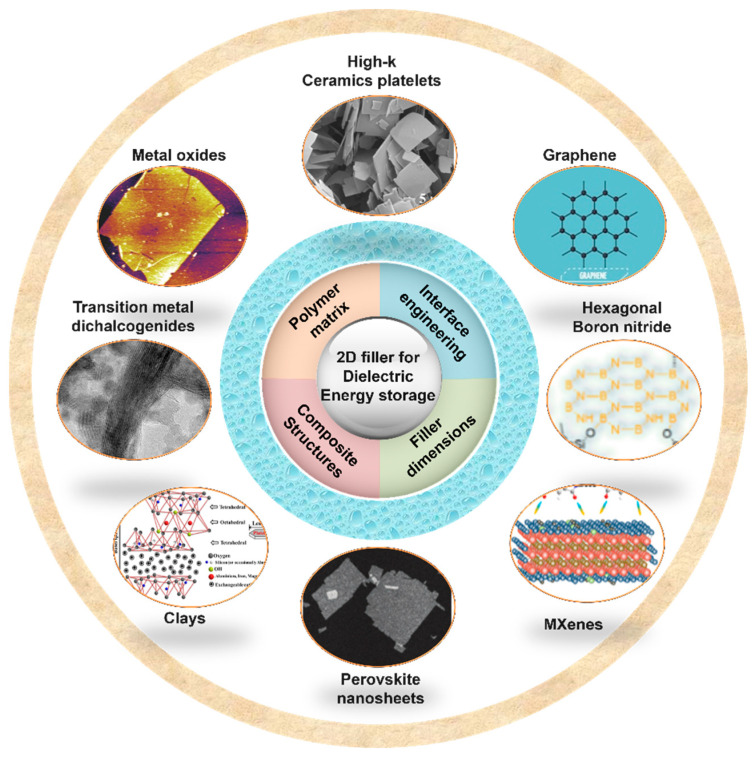
Schematic diagram of the main topics in this review. The polymer matrix, filler dimensions, composite structures, and interface engineering are the main four aspects of polymer-based dielectric composites. Various 2D fillers, including high-k ceramics (reproduced with permission from [20], Copyright 2019, Elsevier Ltd.), perovskite nanosheets (reproduced with permission from [21], Copyright 2020, WILEY-VCH), metal oxides (reproduced with permission from [22], Copyright 2017, WILEY-VCH), graphene (from freepik.com), boron nitride (reproduced with permission from [23], Copyright (2021) Science China Press. Published by Elsevier B.V. and Science China Press), clays (reproduced with permission from [24], Copyright 2016, IOP Publishing Ltd.), transition metal dichalcogenides (reproduced with permission from [25], Copyright (2021) Elsevier), and MXenes (reproduced with permission from [26], Copyright (2022) Elsevier.), have been used to improve the energy storage performance.

**Figure 4 nanomaterials-13-02842-f004:**
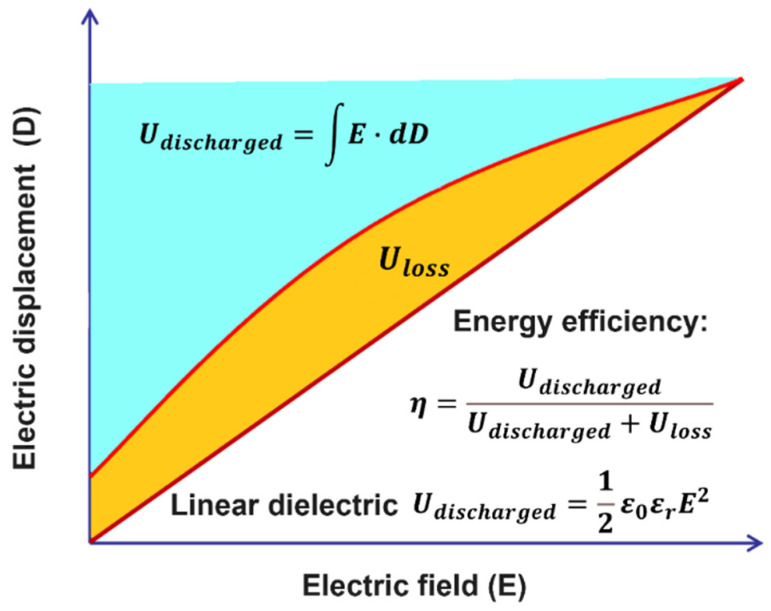
Schematic illustrations D–E are loops of a nonlinear and a linear dielectric.

**Figure 5 nanomaterials-13-02842-f005:**
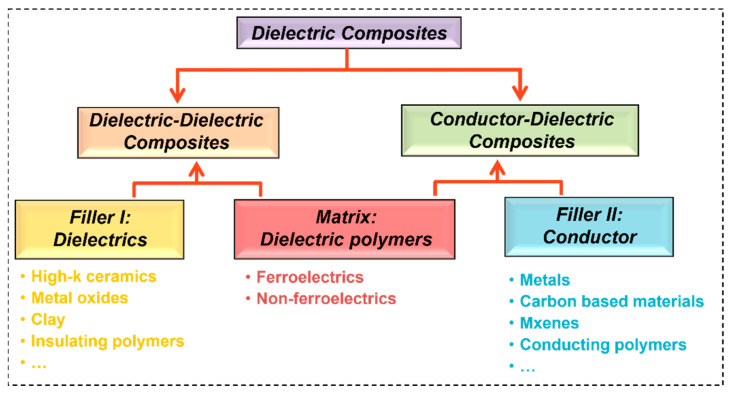
Schematic of two types of polymer-based 0–3 dielectric composites: dielectric–dielectric composites and conductor–dielectric composites.

**Figure 6 nanomaterials-13-02842-f006:**
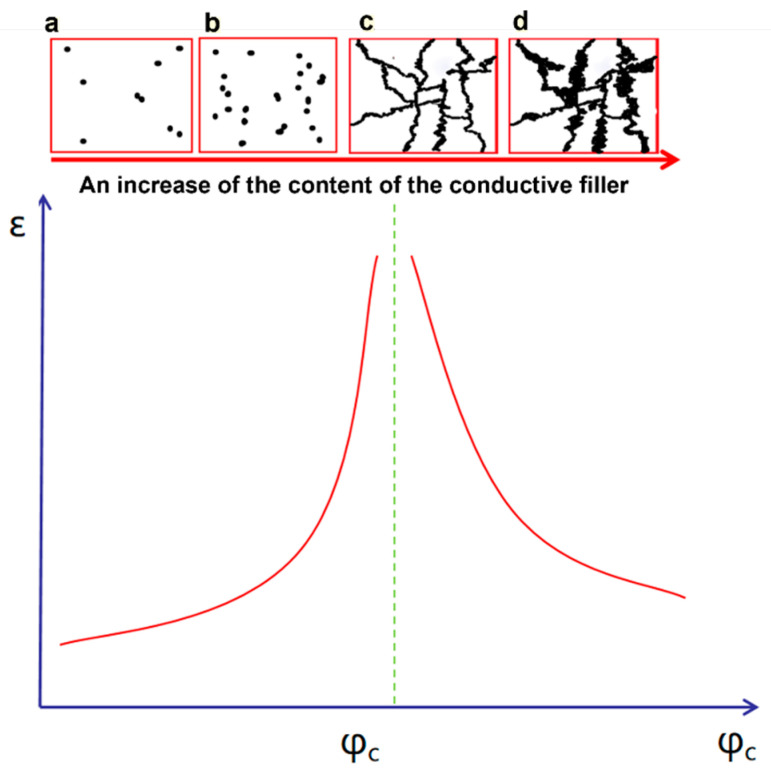
The diagram illustrates the variations in the dielectric permittivity (red solid lines) of composites as a function of concentration, with the dashed green line representing the percolation threshold *φ_c_*. Additionally, the microstructure of the composites is depicted, highlighting the geometric phase transition of the fillers. (**a**) composites with low dielectric permittivity when a low content fillers are in the matrix, (**b**) local clusters of particles begin to form and dielectric permittivity increases, (**c**) conductive particles create infinite conductive cluster in the matrix at a certain filler content (threshold) and the dielectric permittivity reaches the maximum value, and (**d**) more conductive channels form a conductive skeleton and the dielectric permittivity starts decreasing.

**Figure 7 nanomaterials-13-02842-f007:**
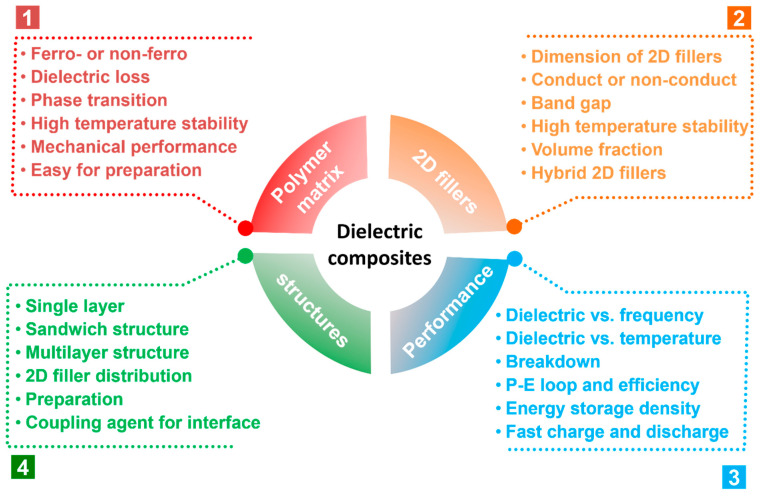
The key points of investigation on polymer-based dielectric composites with 2D fillers.

**Figure 8 nanomaterials-13-02842-f008:**
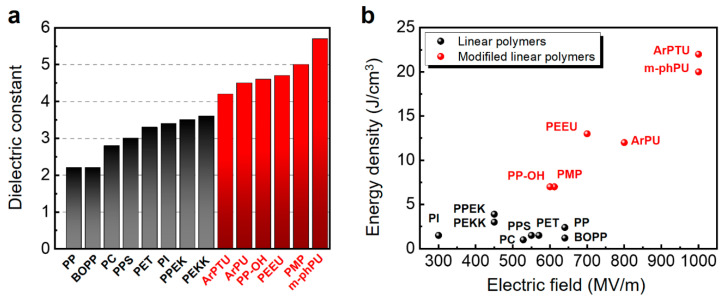
Summary of (**a**) dielectric constant and (**b**) energy density at different electric fields for selected linear polymers and modified linear polymers: polypropylene (PP) [82], polyester (PET) [82], polycarbonate (PC) [82] polyphenylenesulfide(PPS) [82], biaxially oriented polypropylene (BOPP) [83], polyimide (PI) [81], poly(ether ketone ketone) (PEKK) [85], poly(phthalazinone ether ketone) (PPEK) [86], aromatic polyurea (ArPU) [87], aromatic polythiourea (ArPTU) [88], meta-phenylene polyurea (m-phPU) [89], poly(arylene ether urea) (PEEU) [90], modified poly(4-methyl-1-pentene) [91], poly(propylene-co-hexen-6-ol) [92].

**Figure 9 nanomaterials-13-02842-f009:**
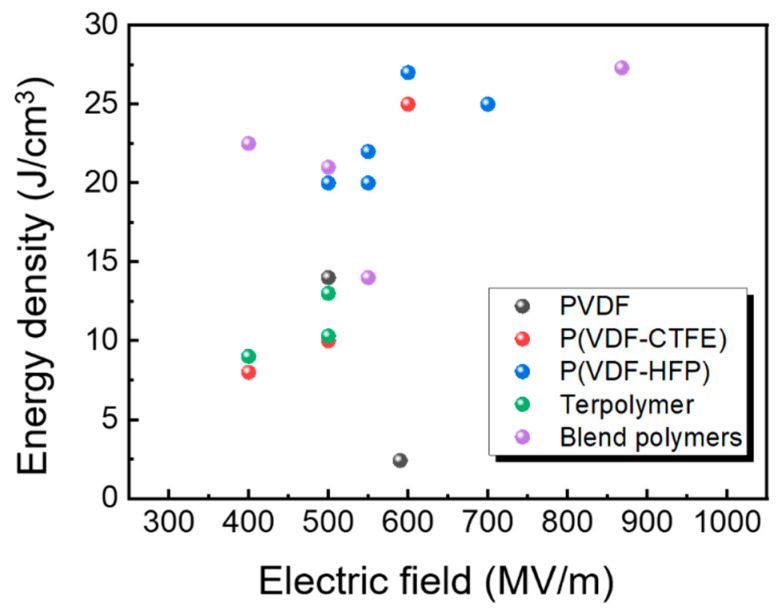
Summary of) energy density at different electric fields for selected linear polymers and modified linear polymers: PVDF [82], quench PVDF [105], dopamine-modified PVDF [106], P(VDF-CTFE)91/9 by extrusion at 190–250 °C [110], P(VDF-CTFE) by quenching in liquid N_2_ [111], P(VDF-CTFE) by cooling to R.T. then annealing at 110 °C [111], P(VDF-HFP) 96/4 by solution casting and uniaxial stretching [107], P(VDF-HFP) 95.5/4.5 by extrusion and stretching at 110 °C [113], P(VDF-HFP) by solution casting [114], P(VDF-HFP) by solution casting, stretching and annealing [114], P(VDF-HFP) by melt-pressing, quenching, and stretching [114], P(VDF-TrFE-CFE) 63/37/7.5 by suspension polymerization [115], P(VDF-TrFE-CTFE) 88.0/5.2/6.8 by direct polymerization [116], P(VDF-TrFE-CTFE) 65.6/26.7/7.7 [117], P(VDF-TrFE-CTFE)-g-PS (14 wt.%) [107], P(VDF-TrFE-CTFE) 80/18/2-g-PEMA (22 wt.%) [118], P(VDF-CTFE) 91/90-BA (10%) [119], α-VDF oligomer-P(VDF) 80/20 [120].

**Figure 10 nanomaterials-13-02842-f010:**
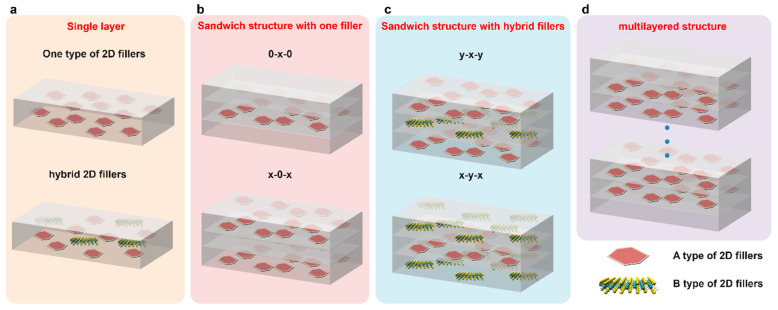
Schematic of different structures of the composites with 2D fillers. (**a**) Single layer, (**b**) sandwich structure with one filler, (**c**) sandwich structure with hybrid fillers, and (**d**) multilayered structure.

**Figure 11 nanomaterials-13-02842-f011:**
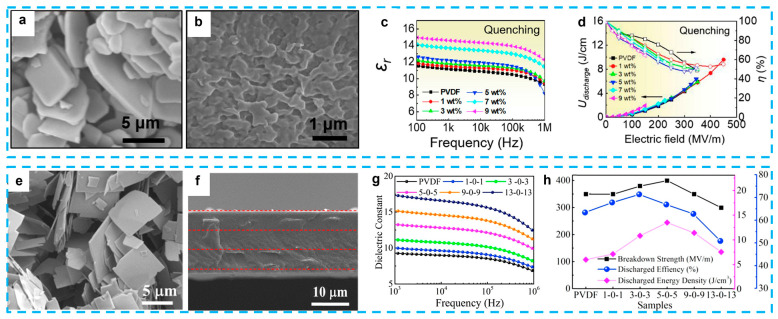
BT/PVDF composites using 2D BT fillers: (**a**) SEM images of BT platelet, (**b**) SEM image of cross-section of BT/PVDF composite film with 1 wt.% of BT, (**c**) the frequency dependence of dielectric constant of composites, and (**d**) the *U_e_* and *η* at different E of all quenching samples. Reproduced with permission from [20], Copyright (2019), Elsevier Ltd. NN/PVDF composites using 2D NN fillers: (**e**) SEM images of 2D NN platelets, (**f**) cross-section of 3-0-3 trilayered architecture composite film, (**g**) the frequency dependence of dielectric constant of composites, and (**h**) discharged energy density of the trilayered architecture composite films and pristine PVDF. Reproduced with permission from [131], Copyright (2017) Elsevier Ltd.

**Figure 12 nanomaterials-13-02842-f012:**
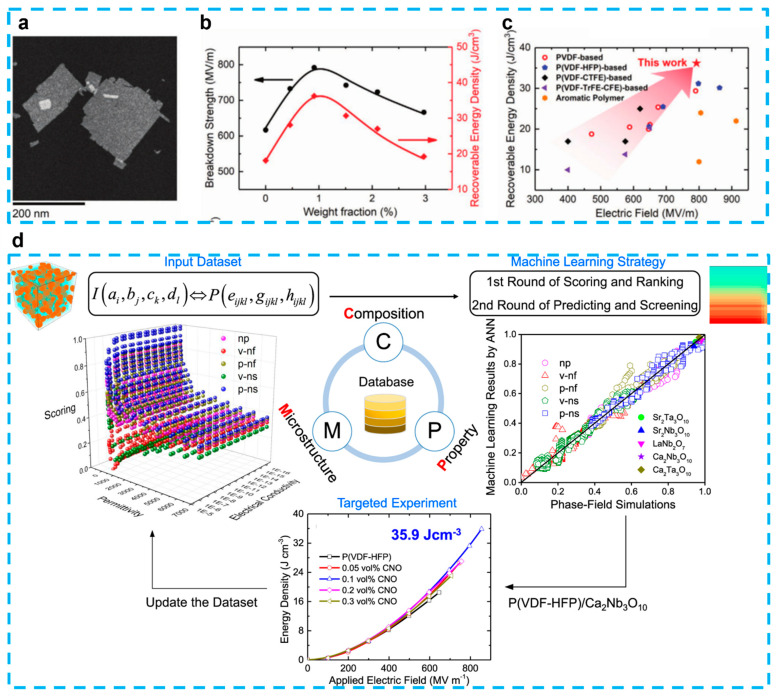
(**a**) TEM image of CNO nanosheets on an ultra-thin copper mesh. (**b**) CNO nanosheet weight fraction dependence of the breakdown strength and the corresponding maximum recoverable energy density. (**c**) Comparisons of the Weibull breakdown strengths and the corresponding maximum recoverable energy densities. Reproduced with permission from [21], Copyright 2020, WILEY-VCH. (**d**) The simulation-guided material development paradigm from theoretical predictions to targeted experiments. Reproduced with permission from [160], Copyright The Authors, 2021, Published by Springer Nature.

**Figure 14 nanomaterials-13-02842-f014:**
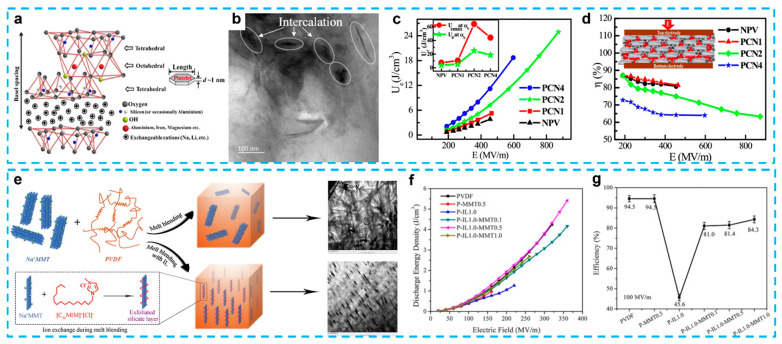
Improved breakdown strength and electrical energy storage performance of PVDF/unmodified montmorillonite clay nano-dielectrics. (**a**) Structure of MMT, 2:1 layered silicate showing two tetrahedral sheets of silicon oxide fused to an octahedral sheet of aluminum hydroxide and platelet structure. (**b**) TEM image of the PCN1 film where intercalation is indicated by marked regions. (**c**) Measured discharged energy density and (**d**) efficiency of all samples. Reproduced with permission from [24], Copyright 2016, IOP Publishing Ltd. Effect of Na^+^ MMT-ionic liquid synergy on electroactive, mechanical, dielectric, and energy storage properties of transparent PVDF-based nanocomposites. (**e**) Schematic representations showing the dispersion states of Na^+^ MMT in PVDF matrix and the mechanism of IL contribute to the exfoliation of Na^+^ MMT, (**f**) discharged energy density, and (**g**) charge–discharge efficiency at 100 MV/m of PVDF and PVDF-based blends. Reproduced with permission from [183], Copyright 2020, Elsevier.

**Figure 15 nanomaterials-13-02842-f015:**
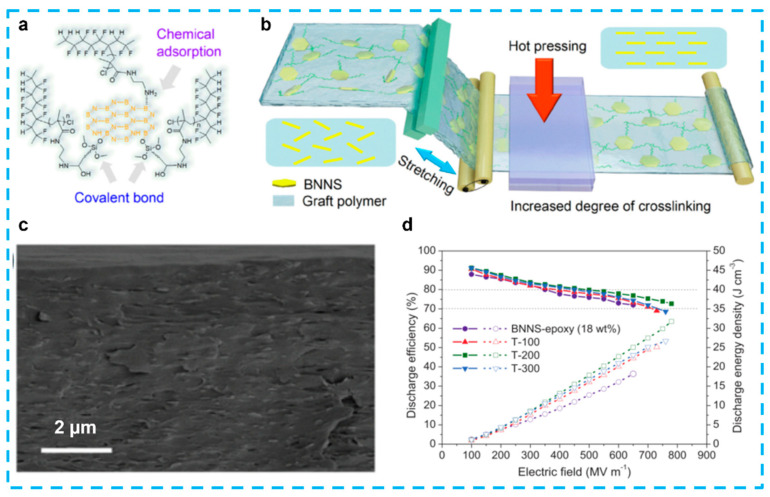
Chemical adsorption on 2D dielectric nanosheets for matrix-free nanocomposites with ultra-high electrical energy storage. (**a**) Schematic diagram of covalent bond and chemical adsorption between polymer chains and fillers. Scheme (**b**) and SEM images (**c**) of oriented epoxy functionalized BNNS after stretching and hot-pressing. (**d**) Discharged density and charge–discharge efficiency of matrix-free nanocomposite film. Reproduced with permission from [23], Copyright (2021) Science China Press. Published by Elsevier B.V. and Science China Press.

**Figure 16 nanomaterials-13-02842-f016:**
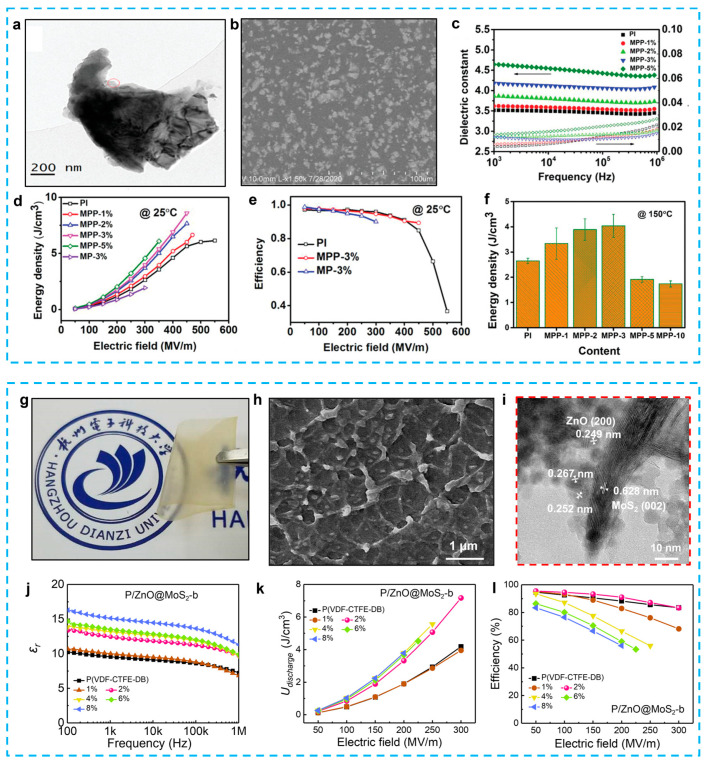
Two dimensional MoS_2_ nanosheet-based polyimide nanocomposite. (**a**) Transmission electron microscopy (TEM) image of the MoS_2_-g-PMMA nanosheets. (**b**) Scanning electron microscopy (SEM) image of the surface morphology. (**c**) Dielectric constant and dielectric loss of nanocomposite films with various MoS_2_ concentrations. (**d**) Energy density of the MoS_2_-g-PMMA/PI nanocomposite films with various concentrations at room temperature. (**e**) Charge–discharging efficiency of the pristine PI, MPP-3%, and MP-3% films at room temperature. (**f**) Comparison of the energy density of the MoS_2_-g-PMMA/PI (where PI is polyimide) nanocomposite with various concentrations at 150 °C. Reproduced with permission from [190]. Copyright (2021), WILEY-VCH. Polymer nanocomposites using hybrid 2D ZnO@MoS_2_ semiconductive nano-fillers. (**g**) Image and (**h**) cross-section SEM images of 2 wt.% P/ZnO@MoS_2_-b composite film. (**i**) HRTEM image of the lattice structure of ZnO@MoS_2_. (**j**) Frequency-dependent dielectric constant of ZnO@MoS_2_ composites samples. (**k**) U_discharge_ and (**l**) *η* at different E for P/ZnO@MoS_2_-b samples. Reproduced with permission from [25], Copyright (2021) Elsevier.

**Figure 17 nanomaterials-13-02842-f017:**
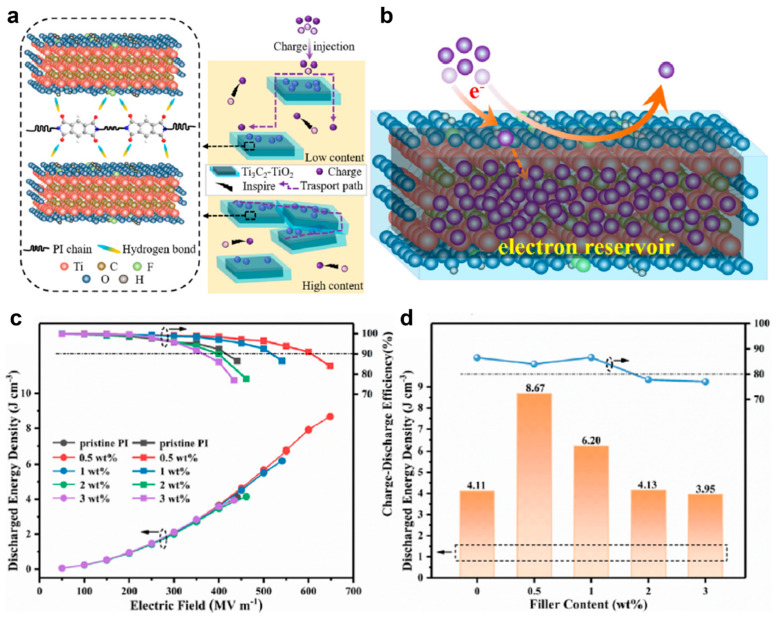
Enhanced breakdown strength and energy density over a broad temperature range in polyimide dielectrics using oxidized MXenes filler. (**a**) Schematic diagram of the fine structure of the polyimide-based nanocomposites and the possible occurrence of the internal charge. (**b**) Discharge energy densities and efficiency of the polyimide-based nanocomposites at different electric fields. (**c**) Variation diagram of maximum discharge energy density and (**d**) efficiency with varied filler contents. Reproduced with permission from [26], Copyright (2022) Elsevier.

## Data Availability

No data availability.

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
