# Peer review of "Energy Storage Performance of Polymer-Based Dielectric Composites with Two-Dimensional Fillers"

_nanomaterials, 2023, doi:10.3390/nano13212842_

Round 1
Reviewer 1 Report
The author describes the “Energy Storage Performance of Polymer based Dielectric Composites with Two-dimensional Fillers”. This Review article is quite interesting from a technological point of view. The author should revise their manuscript based on the comments and suggestions. I recommended a Minor revision of the manuscript.
The Minor suggestion below:
- In the abstract, the author can state a clear research question to convey the main objective of this review?
- The author should elaborate and discuss more detail in the Conductor-dielectric composites in the revised manuscript.
- The author should add more 2D materials based results in the revised manuscript and make the comparison table.
- The introduction is too wide; the author should focus the originality of the current review work.
- The author should improve the grammatical and typo errors in the paper.
- The author should reformat the conclusion remark and the provide narrow conclusion of the paper.
Minor editing of English language required
Reviewer 2 Report
Please find attached the comments.

Moderate editing of English language required
Reviewer 3 Report
This is a comprehensive overview of 2D filler-based composites, encompassing a wide range of materials such as ceramics, metal oxides, carbon compounds, MXenes, clays, boron nitride, etc. The fundamental models for dielectric composites are discussed in Section 2 to show how to predict the dielectric constant with the content of filler. Section 3 focuses on the selection of polymer matrix and how it determines the performance of the composite. Section 4 summarizes the recent progress in achieving enhanced dielectric properties and energy storage capability of 2D polymer nanocomposites, including the structures and various 2D fillers. After that, a summary and some potential outcomes are given. The review is very detailed, with over 250 up-to-date references. Illustrations are fine, helping to follow the discussion. No major remarks for this work. I can only suggest that the authors include a graph on the number of papers on the subject during the last, say, 10 years. The distribution of papers over the countries of the World would also be of interest.
Reviewer 4 Report
The paper is devoted for review energy storage performance of polymer based dielectric composites with two-dimensional fillers. The topic is generally interesting, however the paper contain unexplained places (below) and need major revisions.
Page 9 ‘’and the breakdown strength increased 5%’’, it would be useful to add there the absolute value of the breakdown strength, not only 5%. Fig. 7 it would be useful to add the breakdown strength values there in order to highlight polymers with best properties for energy storage applications.
Page 10, sentence ‘’and have been studied for pyroelectric and piezoelectric applications before, such as aromatic polyuria..’’ I doubt that these polymers have piezoelectric or pyroelectric properties.
sentence ‘’It is necessary to synthesize a polymer possessing two criteria: (i) generate strongly dipolar to enhance dielectric permittivity and (ii) make dipoles following the applied field easily to avoid high loss.’’ Maybe you can explain how it is possible?
In my opinion dipoles always follows the applied field easily when the electromagnetic field frequency is much lower than the reciprocal relaxation time. So, that the main parameter is the relaxation time of dipoles. Please explain which relaxation times values are needed for your applications and how it can be changed.
Fig. 16 should be more commented.
Conclusions should be rewritten in more informative way.
Reviewer 5 Report
The review compiles excellent work in energy storage materials, focusing on polymer matrices and respective fillers. The work is novel, and the authors have made a top-notch quality work with the figures and tables. I especially want to remark on Table 1, an excellent comprehensive summary of polymer energy storage parameters.
To increase the quality of the work, I have added some comments/suggestions for the final version.
- Page 2. Adding precise figures to the referred text will be good for a non-knowledgeable audience. For instance, what are the "wide spectrum of temperatures" and "rapid charge-discharge characteristics"?
- Equation 1 is not mentioned.
- Page 3. The motivation for why there is a "great need for development of new hybrid materials" is unclear. Please expand on this.
- Are the atomic models shown in Figure 3 drawn by the authors and/or from any source? Please specify it.
- Including some microscopies / AFMs of the different fillers will be useful to attract more of the reader's attention. This also includes some examples of the microstructure/morphology for the different polymer-based dielectric composites mentioned in the text.
- Is the percolation behavior shown in Figure 5 applicable to all nanofiller shapes? Would there be any difference if the shapes were particles, rods, or sheets…?
- Interestingly, as pointed out by the authors, most literature reports use thermoplastic polymers for high-temperature film capacitors. Is there any reason behind this in accordance with the authors? If so, I recommend its respective commenting on the text. What about using bio-based or natural polymers such as elastomers containing electrically-conducting nanofillers? (for some reference, please check, e.g., ACS Omega 2019, 4, 2, 3458–3468 and Nature Communications,5, 3132 (2014).
- Is there any practical advantage between 2D fillers, ferroelectric polymers, and non-ferroelectric polymers?
Nothing relevant to add.
Round 2
Reviewer 4 Report
Authors make proper corrections according to reviewer remarks and I suggest to publish the paper as it is.